# Private Set Union with Multiple Contributions

**Travis Dick**
Google Research

**Haim Kaplan**
Tel Aviv University
and Google Research

**Alex Kulesza**
Google Research

**Uri Stemmer**
Tel Aviv University
and Google Research

**Ziteng Sun**
Google Research

**Ananda Theertha Suresh**
Google Research

## Abstract

In the private set union problem each user owns a bag of at most $k$ items (from some large universe of items), and we are interested in computing the union of the items in the bags of all of the users. This is trivial without privacy, but a differentially private algorithm must be careful about reporting items contained in only a small number of bags. We consider differentially private algorithms that always report a subset of the union, and define the utility of an algorithm to be the expected size of the subset that it reports.

Because the achievable utility varies significantly with the dataset, we introduce the *utility ratio*, which normalizes utility by a dataset-specific upper bound and characterizes a mechanism by its lowest normalized utility across all datasets. We then develop algorithms with guaranteed utility ratios and complement them with bounds on the best possible utility ratio. Prior work has shown that a single algorithm can be simultaneously optimal for all datasets when $k = 1$, but we show that instance-optimal algorithms do not exist when $k > 1$, and characterize how performance degrades as $k$ grows. At the same time, we design a private algorithm that achieves the maximum possible utility, regardless of $k$, when the item histogram matches a prior prediction (for instance, from a previous data release) and degrades gracefully with the $\ell_\infty$ distance between the prediction and the actual histogram when the prediction is imperfect.

## 1 Introduction

Consider a dataset where each entry is a set of items donated by a different user. The *set union* problem is to output the union of all of the sets. This simple problem arises in many practical scenarios, and when the items have the potential to be sensitive we may want privacy guarantees to ensure that the result does not reveal personal data. For example, private set union can be used for discovering n-grams in a corpus [Gopi et al., 2020], releasing keys in SQL queries [Wilson et al., 2020], and in general for determining the domain of private aggregate statistics [Amin et al., 2022].

Since the number of conceivable items (e.g., all possible n-grams) can be very large, it is often necessary for the algorithm to restrict its output to a subset of the true union [Gopi et al., 2020, Desfontaines et al., 2022]. Motivated by this, Cohen et al. [2021], Desfontaines et al. [2022] proposed an optimal $(\varepsilon, \delta)$-differentially private algorithm when each user contributes exactly one item. However, in many realistic settings users can contribute multiple items. This prompts a natural question: *can we design an optimal $(\varepsilon, \delta)$-differentially private algorithm when each user contributes up to $k$ items?*

We begin with the definition of differential privacy.

39th Conference on Neural Information Processing Systems (NeurIPS 2025).

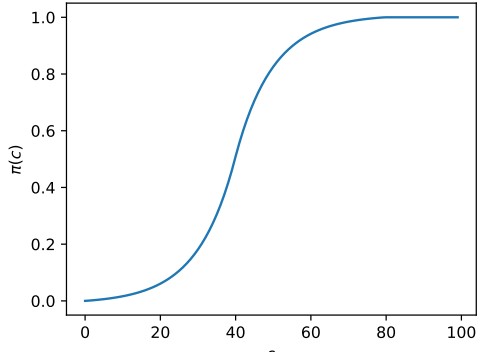

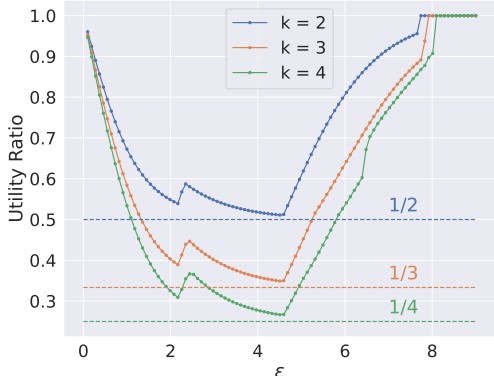

Figure 1: The maximum probability $\pi(c; \varepsilon, \delta)$ of reporting an item with count $c$ when $\varepsilon = 0.1$ and $\delta = 0.0001$.

Figure 2: Optimal utility ratios over datasets with three users for $\delta = 0.01$ and various settings of $\varepsilon$ and $k$. For intermediate values of $\varepsilon$ the ratio can be nearly as low as $1/k$.

**Definition 1.1** (Differential privacy [Dwork et al., 2006])**.** *A randomized algorithm $M$ satisfies $(\varepsilon, \delta)$-differential privacy if for any two neighboring datasets $D, D'$ and for any subset of the output space $\mathcal{S}$, it holds that*

$$Pr[M(D) \in \mathcal{S}] \leq e^\varepsilon \cdot Pr[M(D') \in \mathcal{S}] + \delta.$$

*Two datasets $D$ and $D'$ are neighboring if and only if $d_{\mathrm{ham}}(D, D') \triangleq |D \setminus D'| + |D' \setminus D| = 1$.*

We can now provide a formal definition of the differentially private set union problem.

**Definition 1.2** (Differentially private set union)**.** *Fix a universe $\mathcal{X}$ of items and a contribution bound $k$. Let $D$ be a dataset consisting of bags $B_i \subseteq \mathcal{X}$, $|B_i| \leq k$ for $i \in [n]$. A differentially private set union algorithm $M$ has to output a subset of $\cup_{i \in [n]} B_i$, and its goal is to output a subset which is as large as possible. We denote by $\mathrm{UNION}_k(\varepsilon, \delta)$ the set of all $(\varepsilon, \delta)$-differentially private set union algorithms.*

We define the utility of algorithm $M \in \mathrm{UNION}_k(\varepsilon, \delta)$ on dataset $D$ to be the expected cardinality of its output set, $\mathbb{E}[|M(D)|]$. In the best-case scenario, the utility is equal to the cardinality of the full union, i.e., $\mathbb{E}[|M(D)|] = |\cup_{i \in [n]} B_i|$. However, the best-case utility is typically not achievable. For example, consider a dataset in which each item is contained in the bag of a single user: any algorithm in $\mathrm{UNION}_k(\varepsilon, \delta)$ cannot report more than a $\delta$ fraction of the items in expectation, since for each item there is a neighboring dataset in which it does not exist and hence is reported with probability zero. In general, the achievable fraction of the best-case utility is highly dependent on the item frequencies, making it difficult to compare algorithms across datasets when the goal is a naive maximization of $\mathbb{E}[|M(D)|]$.

The work of Cohen et al. [2021], Desfontaines et al. [2022] suggests an appropriate adjustment. Let the number of times item $x$ appears in dataset $D$ be denoted by $\mathrm{c}(x, D)$ (or simply $\mathrm{c}(x)$ when the underlying dataset is clear). They showed that the utility of any algorithm $M \in \mathrm{UNION}_k(\varepsilon, \delta)$ satisfies

$$\mathbb{E}[|M(D)|] \leq \Pi(D, \varepsilon, \delta) := \sum_{x \in \mathcal{X}} \pi(\mathrm{c}(x); \varepsilon, \delta),$$

where $\pi$ is a sigmoid-like function given by

$$\pi(\mathrm{c}(x); \varepsilon, \delta) = \begin{cases} \frac{e^{\mathrm{c}(x)\varepsilon} - 1}{e^\varepsilon - 1} \cdot \delta & \text{if } \mathrm{c}(x) \leq \mathrm{c}_\ell \\ \left(1 - e^{-(\mathrm{c}(x) - \mathrm{c}_\ell)\varepsilon}\right)\left(1 + \frac{\delta}{e^\varepsilon - 1}\right) + e^{-(\mathrm{c}(x) - \mathrm{c}_\ell)\varepsilon}\pi(\mathrm{c}_\ell) & \text{if } \mathrm{c}_\ell < \mathrm{c}(x) \leq \mathrm{c}_h \\ 1 & \text{otherwise} \end{cases}$$

and $\mathrm{c}_\ell$ and $\mathrm{c}_h$ are constants given by

$$\mathrm{c}_\ell = 1 + \left\lfloor \frac{1}{\varepsilon} \ln\left(\frac{e^\varepsilon + 2\delta - 1}{(e^\varepsilon + 1)\delta}\right) \right\rfloor \qquad \mathrm{c}_h = \mathrm{c}_\ell + \left\lfloor \frac{1}{\varepsilon} \ln\left(1 + \frac{e^\varepsilon - 1}{\delta}(1 - \pi(\mathrm{c}_\ell, \varepsilon, \delta))\right) \right\rfloor. \quad (1)$$

In the special case $\varepsilon = 0$, we have $\pi(c(x); 0, \delta) = \min(c(x)\delta, 1)$. See Figure 1 for an illustration of $\pi$.

Cohen et al. [2021], Desfontaines et al. [2022] showed that the upper bound of $\Pi(D, \varepsilon, \delta)$ is achievable, simultaneously for all datasets, when $k = 1$. We will see later that this is not possible when $k > 1$ (except under certain extreme values of $\varepsilon$ and $\delta$). However, $\Pi(D, \varepsilon, \delta)$ *is* achievable, regardless of $k$, for any *single* dataset (see Theorem 1.3 below). This is not trivial since algorithms in $\text{UNION}_k(\varepsilon, \delta)$ can only return items appearing in their input dataset, which rules out constant algorithms that ignore their input and return a fixed result. (Such algorithms are differentially private and, in other settings, can be used to trivially obtain optimality for any single dataset.)

The achievability of $\Pi(D, \varepsilon, \delta)$ when $k = 1$ motivates its use as a normalizer for the utility $\mathbb{E}[|M(D)|]$. We introduce the following target measure, which we use to establish bounds on the performance of algorithms in $\text{UNION}_k(\varepsilon, \delta)$.

**Definition 1.3** (Utility ratio). *The* utility ratio *of an algorithm* $M \in \text{UNION}_k(\varepsilon, \delta)$ *is*

$$u_k(M) := \min_{D \in \mathcal{D}_k} \frac{E_M[|M(D)|]}{\Pi(D, \varepsilon, \delta)} \ ,$$

*where $\mathcal{D}_k$ is the collection of all nonempty datasets where each user contributes at most $k$ items.*

That $u_k(M)$ is generally less than one when $k > 1$ is easily demonstrated numerically using a linear program that finds the optimal mechanism for a finite collection of datasets. Figure 2 shows that the resulting behavior is complex, even considering only very small datasets, and the worst-case utility ratio appears to be close to $1/k$. Our aim is to characterize this behavior theoretically.

## 1.1 Main results

**Theorem 1.1** (Informal impossibility results). *Let $\delta = O_\varepsilon(1/k^2)$, where the subscript denotes an unstated dependence on $\varepsilon$. Then for any algorithm $M$ in $\text{UNION}_k(\varepsilon, \delta)$ we have*

$$u_k(M) = O\left(\frac{1}{k}\left(1 + \frac{\ln k}{\varepsilon}\right)\right). \qquad \text{(Theorem 2.2)}$$

*In addition, even if $D$ is restricted to "easy" datasets where $\Pi(D, \varepsilon, \delta) = \Omega(|\mathcal{X}|)$, we still have*

$$u_k(M) = \tilde{O}_n\left(\frac{1}{k^{1/4}}\right), \qquad \text{(Theorem 2.3)}$$

*where $\tilde{O}$ hides logarithmic terms.*

Theorem 1.1 shows that instance-optimal algorithms are not generally possible when $k > 1$, with the bounds roughly matching the minimums in Figure 2. However, we can still construct algorithms with meaningful utility guarantees.

**Theorem 1.2** (Informal achievability results). *There exists an algorithm $M$ in $\text{UNION}_k(\varepsilon, \delta)$ such that for every dataset $D$,*

$$\mathbb{E}[|M(D)|] = \frac{1}{k} \cdot \Pi(D, \varepsilon', \delta'),$$

*where $\varepsilon' = \tilde{\Omega}(\varepsilon)$ and $\delta' = \tilde{\Omega}(\delta/e^\varepsilon)$ (see Theorem 3.2). In addition, when $\varepsilon = 0$ or $\varepsilon \to \infty$ (holding $\delta$ constant), there exists an $M$ such that $u_k(M) = 1$ (see Lemmas 3.5 and 3.6).*

The results above raise an important question: can we *ever* do better than a utility ratio of $O(1/k)$? Theorem 1.3 shows that, if we can predict the histogram of items in the dataset in advance, then there exists a private set union algorithm achieving the optimal utility regardless of $k$.

**Theorem 1.3** (Informal achievability with predictions). *For any nonempty histogram over $\mathcal{X}$ and privacy parameters $\varepsilon$ and $\delta$, there exists an algorithm $M \in \text{UNION}_k(\varepsilon, \delta)$ such that*

$$\mathbb{E}\left[|M(D)|\right] = \Pi(D, \varepsilon, \delta)$$

*for any $D$ matching the predicted histogram, regardless of the contribution bound $k$ (Theorem 4.1).*

The private algorithm $M$ satisfying Theorem 1.3 also performs well on datasets that are "similar" to the target dataset $D$, making it an appropriate algorithm for settings where some public prediction regarding the union is available (see Section 4 for more details).

**A note about $k$**: In some settings we may not have any *a priori* contribution bound $k$, in which case we need to choose one and enforce it. This introduces a natural tradeoff: larger $k$ retain more data, but reduce the utility ratio (as indicated by our results). Similar contribution-bounding tradeoffs have been explored in prior work [Amin et al., 2019, Epasto et al., 2020, Amin et al., 2022]. Although it is not our main focus here, in Appendix A we show one way that $k$ can be selected privately when no contribution bound is known in advance.

## 1.2 Related work

The differentially private set union problem was implicitly introduced by Korolova et al. [2009] in the early days of differential privacy. Subsequent work by Gopi et al. [2020], Carvalho et al. [2022] improved utility by processing users sequentially and choosing contributions in a clever way, minimizing waste on heavy items while maintaining low sensitivity. Swanberg et al. [2023], Chen et al. [2024] proposed multi-round mechanisms with careful budget-splitting across the rounds; this allowed them to process users in parallel while retaining good utility. However, none of these works provide worst case utility guarantees, and they primarily compare different approaches empirically.

The optimal reporting probabilities when $k = 1$ were introduced by Desfontaines et al. [2022], Cohen et al. [2021]. [Knop and Steinke, 2023] studied the related problem of estimating the *size* of the union rather than the union itself. More distantly related work on privately finding the $k$ most frequent items in a database was published by Bhaskar et al. [2010], Durfee and Rogers [2019], McKenna and Sheldon [2020], Gillenwater et al. [2022].

## 2 Impossibility results

In this section we derive upper bounds on the utility ratio of any $(\varepsilon, \delta)$-differentially private set union mechanism when applied to datasets with contribution bound $k$. Our upper bounds all diminish with $k$, showing that for datasets with large contribution bounds, no differentially private mechanisms can meaningfully compete with the optimal utility simultaneously for all datasets. All omitted proofs are given in Appendix B.

Our first result shows that there exist regimes for $\varepsilon$ and $\delta$ such that every set union mechanism has utility ratio $O(1/k)$. In particular, as the contribution bound $k$ grows, no mechanism is competitive with $\Pi(D)$ on every dataset $D$, despite the fact that Theorem 1.3 establishes a mechanism matching $\Pi(D)$ for any single $D$.

**Theorem 2.1** (Warm-up)**.** *Let $k \geq 2$, $\varepsilon \geq 0$, $\delta \leq \frac{1}{e^\varepsilon + 2}$, and let $M$ be any $(\varepsilon, \delta)$-differentially private set union mechanism. Then there exists a dataset $D$ with contribution bound $k$ such that*

$$\frac{\mathbb{E}\big[|M(D)|\big]}{\Pi(D, \varepsilon, \delta)} \leq \frac{1}{k} + \frac{k}{e^\varepsilon + k}.$$

*In particular, for $\varepsilon = 2\ln(k)$ and $\delta \leq \frac{1}{k^2 + 2}$, we have $\frac{\mathbb{E}[|M(D)|]}{\Pi(D, \varepsilon, \delta)} \leq \frac{2}{k}$.*

*Proof sketch.* Let the item universe $\mathcal{X}$ contain $k$ items, and let $D$ be the dataset with a single user that contributes every item, i.e., $B_1 = \mathcal{X}$. For each item $x \in \mathcal{X}$, construct a dataset $D_x$ by adding a second user to $D$ that contributes only item $x$, i.e., $B_2 = \{x\}$. The privacy parameters $\varepsilon$ and $\delta$ are chosen to ensure that $\pi(2)$ is much larger than $\pi(1)$, so a mechanism $M$ is only competitive on $D_x$ if it outputs item $x$ with probability close to $\pi(2)$. However, since $D_x$ neighbors a dataset containing only item $x$ (after removing user 1), the total probability mass of outputting any set containing an item other than $x$ must be at most $\delta$. Thus, $\Pr\big(x \in M(D_x)\big) \leq \Pr\big(M(D_x) = \{x\}\big) + \delta$. And, since $M$ is DP, we further have $\Pr\big(x \in M(D_x)\big) \leq e^\varepsilon \Pr\big(M(D) = \{x\}\big) + 2\delta$. So, for $M$ to compete with $\Pi(D_x)$, we require that $M$ outputs the singleton set $\{x\}$ with non-trivial probability when run on the single-user dataset $D$. On the other hand, $D$ neighbors the empty dataset, so the total probability mass it assigns to non-empty outputs is at most $\delta$, implying that there exists an item $y$ such that $\Pr\big(M(D) = \{y\}\big) \leq \delta/k$. Therefore, $\Pr\big(y \in M(D_y)\big) \leq e^\varepsilon \delta/k + 2\delta$. By contrast, in the specified

parameter regime we have $\pi(2) = \delta e^\varepsilon + \delta$. When $\varepsilon$ is sufficiently large that the $e^\varepsilon \delta$ terms dominates, the mechanism $M$ is only able to output $y$ with probability approximately $\pi(2)/k$.

Intuitively, the only way for $M$ to compete with $\Pi(D_x)$ is for $M$ to have an output distribution on the single-user dataset $D$ that prioritizes $x$, and it is not possible for a single mechanism $M$ to do this simultaneously for all $x \in \mathcal{X}$. $\square$

A weakness of Theorem 2.1 is that the utility ratio bound of $2/k$ holds only when $\varepsilon \geq 2 \ln k$. For large $k$, this is an extremely low privacy regime. The next result extends the argument of Theorem 2.1 and establishes a bound of $O(\frac{\ln k}{k \varepsilon})$ on the utility ratio of any mechanism that holds in almost any privacy regime. In particular, it holds for any $\varepsilon$ as long as $\delta$ decays like $1/k^2$.

**Theorem 2.2.** *Let $k \geq 4$, $\varepsilon \geq 0$, $\delta \leq \frac{1}{k^2} \cdot \frac{1}{e^{\varepsilon/2}} \cdot \frac{e^\varepsilon - 1}{e^\varepsilon + 1}$ and $M$ be any $(\varepsilon, \delta)$-differentially private set union mechanism. Then there exists a dataset $D$ with contribution bound $k$ such that*

$$\frac{\mathbb{E}[|M(D)|]}{\Pi(D, \varepsilon, \delta)} \leq \frac{12}{k-1}\left(1 + \frac{1}{\varepsilon}\log k\right).$$

*Proof sketch.* The proof follows a similar argument to the one for Theorem 2.1, but instead of adding a single user to $D$, we add $O(\ln(k))$ users, each contributing a constant fraction of the previous user's items. The key advantage of this iterative construction is that the suboptimality incurred by the mechanism is determined by its inability to output items with sufficiently large probability across a range of item counts from 1 to $\ln(k)$. In particular, rather than requiring $\varepsilon$ to be $O(\ln(k))$ to ensure that $\pi(2)$ is much larger than $\pi(1)$, here we allow for constant $\varepsilon$ and drive suboptimality from the ratio between $\pi(\ln(k))$ and $\pi(1)$. $\square$

The datasets that witness the utility ratio upper bound in Theorem 2.2 have the property that $\Pi(D, \varepsilon, \delta)/|\cup_{i \in [n]} B_i|$ tends to zero as the contribution bound $k$ grows. In other words, even the optimal mechanisms for datasets $D$ established by Theorem 1.3 are only able to return a vanishing fraction of the items contained in $D$ as the contribution bound $k$ grows. Our final impossibility result shows that even on a class of datasets where $\Pi(D) \geq |\cup_{i \in [n]} B_i|/2$, the utility ratio achievable by any mechanism diminishes with $k$, albeit at a slower rate than for the previous results.

**Theorem 2.3.** *Let $k \geq 2$, $n > 2 \cdot c_h$ (so that $\pi(n/2) = 1$), $\varepsilon \geq 1/n$, and $\delta < 1/(40 e^\varepsilon n^{1/2} k^{1/4})$. Let $M$ be any $(\varepsilon, \delta)$-differentially private set union mechanism. Then there exists a dataset $D$ with contribution bound $k$ such that*

$$\frac{\mathbb{E}[|M(D)|]}{\Pi(D, \varepsilon, \delta)} = O\left(\left(\frac{n^2 \log(nk)}{k}\right)^{1/4}\right).$$

*Proof sketch.* The key idea is a reduction showing that a private set union mechanism $M$ can be used to construct a mechanism for estimating matrix marginals whose performance is related to the utility ratio $u_k(M)$. Combined with an impossibility result for privately estimating matrix marginals based on robust fingerprinting codes (modified from the work of Steinke and Ullman [2015]), this yields a bound on the utility ratio.

The marginal problem we reduce from is the following: given a binary matrix $C \in \{0, 1\}^{n \times k}$, the mechanism aims to output a vector in $\{0, 1\}^k$ such that whenever a column of $C$ is entirely 0 or 1, the corresponding component of the output vector is also equal to 0 or 1 (respectively). On mixed columns of $C$, the mechanism can output either 0 or 1. This is an easier problem than computing the column marginals of $C$ (i.e., the fraction of 1s per column), since the mechanism is only required to identify "pure" columns. We are interested in mechanisms that are at most $(\beta, \gamma)$-inaccurate, which requires that with probability at least $\gamma$ their output is correct on all but at most $\beta k$ columns. Steinke and Ullman [2015] upper bound $\beta$ for differentially private mechanisms.

The reduction works as follows: view row $i$ of the matrix $C$ as the indicator vector for user $u_i$'s bag of items from a universe of size $k$. Given a mechanism $M$ for private set union, we obtain a set $\hat{U}$ approximating the union of the contributed items. We then output the vector $\hat{m} \in \{0, 1\}^k$ where $\hat{m}_j = 1$ if $j \in \hat{U}$ and $\hat{m}_j \sim \text{Bernoulli}(1/2)$ if $j \notin \hat{U}$. Because $M \in \text{UNION}_k(\varepsilon, \delta)$, every index $j \in \hat{U}$ *must* be a column of $C$ that contains at least one 1, so we never make mistakes on those columns. And for each pure column $j \notin \hat{U}$, we have a 1/2 chance of correctly guessing whether the

column was all 0s or all 1s. It follows that the expected number of mistakes made by the reduction mechanism is at most $(k - u_k(M) \cdot \Pi(D))/2$, where $D$ is the set union instance encoded by the rows of $C$. To finish the proof, we construct $C$ to ensure that $\Pi(D) \geq k/2$, which ensures the expected fraction of marginal mistakes is bounded in terms of $u_k(M)$. Then we convert this to a high probability bound that contradicts the impossibility result for the marginal problem when $u_k(M)$ is too large. □

# 3 Algorithms with utility guarantees

## 3.1 A simple budget splitting algorithm

A straightforward approach when $k > 1$ is to divide the budget by $k$ and apply the optimal $k = 1$ algorithm, including each item $x \in \mathcal{X}$ in the output independently with probability $\pi(\mathrm{c}(x, D); \varepsilon/k, \delta/k)$. Clearly the utility of this mechanism is $\Pi(D, \varepsilon/k, \delta/k)$. The following lemma argues that it is private. The proof is straightforward and included for completeness in Appendix C.

**Lemma 3.1.** *Let $M_{split}(D; \varepsilon, \delta, k)$ be the mechanism that works as follows: for each item $x \in \mathcal{X}$, include $x$ in the output with probability $\pi(\mathrm{c}(x, D); \varepsilon/k, \delta/k)$. Then $M_{split}$ is a $(\varepsilon, \delta)$-differentially private set union mechanism when users contribute at most $k$ items.*

## 3.2 Bicriteria Approximation

The simple budget splitting algorithm achieves the $\Pi(D, \varepsilon, \delta)$ bound of Theorem 1.3 for every dataset $D$ but with privacy parameters smaller by a factor of $k$ than the target parameters. In the following theorem we compete with this bound for a larger value of $\varepsilon$, smaller than the "real" $\varepsilon$ by only a factor of $\ln(1/\delta)$. This gain comes with a multiplicative loss of $1/k$ over the $\Pi(D, \varepsilon, \delta)$ bound.

**Theorem 3.2.** *Let $\varepsilon, \delta < 1$ be small enough constants. There exists an $(\varepsilon, \delta)$-DP algorithm whose expected number of identified items is*

$$\frac{1}{k} \cdot \Pi \left( D, \Omega \left( \frac{\varepsilon}{\ln(1/\delta)} \right), \Omega \left( \frac{\delta}{\ln(1/\delta)e^\varepsilon} \right) \right).$$

We refer to this result as "bicriteria" because our $(\varepsilon, \delta)$-DP algorithm incurs a multiplicative loss of $\frac{1}{k}$ when compared not with the optimal reporting probabilities for parameters $(\varepsilon, \delta)$, but rather with those for the relaxed parameters $\left( \frac{\varepsilon}{\ln(1/\delta)}, \frac{\delta}{\ln(1/\delta)e^\varepsilon} \right)$.

Our bicriteria algorithm, called `Bicrit`, is given below. We present an alternative construction of a bicriteria algorithm in Appendix C.1.

---

**Algorithm 1** `Bicrit`

---

**Notation:** Let $k$ denote the contribution bound, let $\mathcal{X}$ be a domain of items, and let $\Delta_{\mathcal{X},k} = \{B \subseteq \mathcal{X} : |B| \leq k\}$ denote the set of all possible bags of size at most $k$ from $\mathcal{X}$.

**Input:** Dataset $D \in (\Delta_{\mathcal{X},k})^n$ containing $n$ bags, privacy parameters $\varepsilon, \delta > 0$.

1. Denote $\hat{\varepsilon} = \frac{\varepsilon}{4 \ln(2/\delta)}$ and $\hat{\delta} = \frac{\delta}{8 \log(2/\delta)e^\varepsilon}$

2. For each $x \in \mathcal{X}$:
   (a) Let $b_x \leftarrow \text{Bernoulli} \left( \frac{1}{k} \right)$
   (b) If $b_x = 1$ then report $x$ with probability $\pi(\mathrm{c}(x); \hat{\varepsilon}, \hat{\delta})$

---

Note that Algorithm `Bicrit` does not need to explicitly traverse all $x \in \mathcal{X}$; we can skip items to which no user contributes since $\pi(0; \hat{\varepsilon}, \hat{\delta}) = 0$.

The next lemma captures the privacy guarantee of Algorithm `Bicrit`.

**Lemma 3.3.** *Algorithm `Bicrit` is $(\varepsilon, \delta)$-DP.*

*Proof.* Fix two neighboring datasets $D^0$ and $D^1 = D^0 \cup \{B\}$ for $B = \{x_1, x_2, \ldots, x_z\}$ where $z \leq k$. Let $\ell = |\{x \in B : b_x = 1\}|$ be the random variable denoting the number of elements from $B$

that are sampled in Step 2a. Let $E$ denote the event that $\ell \leq \ell_0 := 4\ln(2/\delta)$, and $\bar{E}$ its complement. By the Chernoff bound we have $\Pr\left[\bar{E}\right] \leq \delta/2$. Now, by composition (and by our choice of $\hat{\varepsilon}$ and $\hat{\delta}$ in Step 1), for any outcome event $F$ we have that

$$\Pr[\mathtt{Bicrit}(D^0) \in F] = \Pr[E] \cdot \Pr[\mathtt{Bicrit}(D^0) \in F|E] + \Pr\left[\bar{E}\right] \cdot \Pr\left[\mathtt{Bicrit}(D^0) \in F|\bar{E}\right]$$

$$\leq \Pr[E]\left(e^{\hat{\varepsilon}\ell_0} \cdot \Pr[\mathtt{Bicrit}(D^1) \in F|E] + \ell_0 e^{(\ell_0 - 1)\hat{\varepsilon}}\hat{\delta}\right) + \frac{\delta}{2}$$

$$\leq \Pr[E]\left(e^{\varepsilon} \cdot \Pr[\mathtt{Bicrit}(D^1) \in F|E] + \frac{\delta}{2}\right) + \frac{\delta}{2}$$

$$\leq e^{\varepsilon} \cdot \Pr[\mathtt{Bicrit}(D^1) \in F] + \delta.$$

$\square$

The utility analysis of the bicriteria algorithm is straightforward:

**Lemma 3.4.** *The expected number of identified items in Algorithm* `Bicrit` *is*

$$\frac{1}{k} \cdot \Pi\left(D, \Omega\left(\frac{\varepsilon}{\ln(1/\delta)}\right), \Omega\left(\frac{\delta}{\ln(1/\delta)e^{\varepsilon}}\right)\right).$$

*Proof.* For any dataset $D$ we have

$$\mathbb{E}[|\mathtt{Bicrit}(D)|] = \sum_{x \in \mathcal{X}} \frac{1}{k} \cdot \pi(\mathrm{c}(x); \hat{\varepsilon}, \hat{\delta}) = \frac{1}{k} \cdot \Pi\left(D, \hat{\varepsilon}, \hat{\delta}\right)$$

$$= \frac{1}{k} \cdot \Pi\left(D, \frac{\varepsilon}{4\ln(2/\delta)}, \frac{\delta}{8\ln(2/\delta)e^{\varepsilon}}\right).$$

$\square$

### 3.3 Optimal Mechanisms in Extreme Privacy Regimes

Finally, we describe some mechanisms that behave optimally when the privacy parameter $\varepsilon$ is extremely large or small.

**Small $\varepsilon$ regime.** When $\varepsilon = 0$ the $\pi$ function takes a particularly simple form: $\pi(c; 0, \delta) = \min(c\delta, 1)$. The following lemma gives a mechanism $M_u$ that matches these output probabilities as long as $\delta < 1/n$, where $n$ is the number of users. Its proof is straightforward and omitted.

**Lemma 3.5.** *Let $n$ be the number of users and assume that $\delta < 1/n$. Let $M_0(D; \delta)$ be a mechanism that with probability $n\delta$ picks a user $i$ uniformly at random and outputs the set of items in $B_i$ and with probability $1 - n\delta$ outputs the empty set. Then $M_0$ is $(0, \delta)$ differentially private, and it outputs each item $x$ with probability $\mathrm{c}(x, D) \cdot \delta = \pi(\mathrm{c}(x, D); 0, \delta)$.*

**Large $\varepsilon$ regime.** We describe a mechanism that achieves the optimal utility $\Pi(D, \varepsilon, \delta)$ as $\varepsilon \to \infty$. The mechanism composes the budget splitting mechanism of Section 3.1 with a simple mechanism $M_{\mathrm{all}}$ that outputs the full union with probability $\delta$ and otherwise outputs the empty set. Importantly, $M_{\mathrm{all}}$ outputs items that appear exactly once with the maximum possible probability $\pi(1; \varepsilon, \delta) = \delta$. Lemma C.7 in Appendix C.2 shows that $M_{\mathrm{all}}$ is a $(0, \delta)$-differentially private.

The intuition underlying the combination of $M_{\mathrm{all}}$ and $M_{\mathrm{split}}$ is as follows. For any dataset $D$, $M_{\mathrm{split}}$ outputs each item with probability $\pi(\mathrm{c}(x, D); \varepsilon/k, \delta/k)$ which is smaller than $\pi(\mathrm{c}(x, D); \varepsilon, \delta)$. However, for all items that appear at least twice, both probabilities converge to 1 in the limit as $\varepsilon \to \infty$. The only catch is that $M_{\mathrm{split}}$ outputs items appearing exactly once with probability $\delta/k$ instead of $\delta$ (regardless of $\varepsilon$). To fix this, we compose $M_{\mathrm{all}}$ and $M_{\mathrm{split}}$, spending most of our $\delta$ budget on $M_{\mathrm{all}}$ to get the maximum output probabilities for items that appear once, and relying on the fact that for any nonzero $\delta$ and count $c \geq 2$, we have $\lim_{\varepsilon \to \infty} \pi(c; \varepsilon, \delta) = 1$. The final mechanism and its properties are summarized in the following lemma, which we prove in Appendix C.2.

**Lemma 3.6.** *Let $M_{large}(D; \varepsilon, \delta, k)$ be the following mechanism: let $\delta' = \delta - \min(\delta, 1/\varepsilon)$ and output the union of $M_{all}(D; \delta')$ and $M_{split}(D; \varepsilon, \delta - \delta', k)$. Then $M_{large}$ is an $(\varepsilon, \delta)$-differentially private set union mechanism. Furthermore, for any contribution bound $k$, dataset $D$ with contributions bounded by $k$, and privacy parameter $\delta$, we have that*

$$\lim_{\varepsilon \to \infty} \frac{\mathbb{E}[|M_{large}(D; \varepsilon, \delta, k)|]}{\Pi(D; \varepsilon, \delta)} = 1.$$

## 4 Leveraging a prediction

Finally, in this section, we study whether predicted information about the underlying dataset $D$, e.g., based on historical runs, can improve the utility for private set union algorithms. In particular, we consider the case where a predicted histogram $H$ for the item counts is available. Our goal is to perform well on datasets whose histogram is close to $H$.

The requirement in Definition 1.2 that the algorithm must output a subset of the input dataset excludes the trivially successful algorithm that always outputs the union $\{x \mid H(x) > 0\}$. However, somewhat surprisingly, we show that if the predicted histogram $H$ is correct, it is possible to design a private set union algorithm $M_H$ that achieves the best possible expected utility $\Pi(D, \varepsilon, \delta)$, regardless of the contribution bound $k$. For any dataset $D$, let $H_D$ be its histogram where $\forall x, H_D(x) = \mathrm{c}(D, x)$. Note that the optimal utility bound $\Pi(D, \varepsilon, \delta)$ only depends on $H_D$. We abuse notation and define

$$\Pi(H, \varepsilon, \delta) = \sum_{x \in \mathcal{X}} \pi(H(x); \varepsilon, \delta),$$

and we have $\forall D, \Pi(D, \varepsilon, \delta) = \Pi(H_D, \varepsilon, \delta)$. Moreover, for any $d > 0$ and histogram $H$, we define

$$\Pi_{-d}(H, \varepsilon, \delta) := \sum_{x \in \mathcal{X}} \pi(H(x) - d; \varepsilon, \delta)$$

to be the $\Pi$ bound when all item counts have been reduced by $d$. The result is stated below.

**Theorem 4.1.** *Let $H$ be a predicted histogram. Then there exists an $(\varepsilon, \delta)$-private set union mechanism $M_H$ such that*

$$\mathbb{E}[|M_H(D)|] = \Pi_{-\ell_\infty(H_D, H)}(H, \varepsilon, \delta),$$

*where in particular we have $\mathbb{E}[|M_H(D)|] = \Pi(D, \varepsilon, \delta)$ if $H_D = H$.*

*Proof.* Given a predicted histogram $H$, we construct the mechanism $M_H$ as follows. Compute $d = \ell_\infty(H_D, H) = \max_x |H_D(x) - H(x)|$ and sample $p \sim U(0, 1)$. Then $M_H$ outputs the set

$$M_H(D) = \{x \mid \pi(H(x) - d; \varepsilon, \delta) > p\}.$$

The utility guarantee follows by noting that

$$\mathbb{E}[|M(D)|] = \sum_{x \in \mathcal{X}} \Pr\big(p < \pi(H(x) - d; \varepsilon, \delta)\big) = \sum_{x \in \mathcal{X}} \pi(H(x) - d; \varepsilon, \delta) = \Pi_{-\ell_\infty(H_D, H)}(H, \varepsilon, \delta).$$

It remains to prove that the algorithm is private. Since $\pi(\mathrm{c} - d; \varepsilon, \delta)$ is a monotonically increasing function of $\mathrm{c}$, the output of the algorithm is determined by $\mathrm{c}_D$, defined as the smallest $\mathrm{c}$ such that $\pi(\mathrm{c} - d; \varepsilon, \delta) \geq p$. To see this, note that we can get $M_H(D)$ by post-processing $\mathrm{c}_D$ and outputting the set $\{x \mid H(x) > \mathrm{c}_D\}$. Hence it is sufficient to prove that $\mathrm{c}_D$ is a private statistic of $D$.

Note that $\mathrm{c}_D$ only depends on $D$ through $d = \ell_\infty(H_D, H)$, and by the definition of $\mathrm{c}_D$, we have

$$\Pr(\mathrm{c}_D = m) = \pi(m - d) - \pi(m - d - 1).$$

We denote the distribution of $\mathrm{c}_D$ when $d = \ell_\infty(H_D, H)$ as $P_d$. For all neighboring datasets $D$ and $D'$, by the reverse triangle inequality we have

$$|\ell_\infty(H_D, H) - \ell_\infty(H_{D'}, H)| \leq \ell_\infty(H_D, H_{D'}) \leq 1.$$

Hence it is sufficient to prove that $\forall d \geq 0$, $P_d$ and $P_{d+1}$ are $(\varepsilon, \delta)$-indistinguishable. More precisely, we want to prove that for $d \geq 0$, we have

$$\Pr_{m \sim P_d}\left(P_d(m) \leq e^\varepsilon P_{d+1}(m)\right) \geq 1 - \delta \tag{2}$$

and

$$\Pr_{m \sim P_{d+1}} \left( P_{d+1}(m) \leq e^\varepsilon P_d(m) \right) \geq 1 - \delta. \tag{3}$$

By the definition of $P_d$, for all $m' \geq 0$, we have

$$P_{d+m'}(m + m') = P_d(m).$$

This implies

$$\Pr_{m \sim P_d} \left( P_d(m) \leq e^\varepsilon P_{d+1}(m) \right) = \Pr_{m \sim P_0} \left( P_0(m - d) \leq e^\varepsilon P_1(m - d) \right)$$

and

$$\Pr_{m \sim P_1} \left( P_1(m - d) \leq e^\varepsilon P_0(m - d) \right) = \Pr_{m \sim P_1} \left( P_1(m - d) \leq e^\varepsilon P_0(m - d) \right).$$

Hence it is sufficient to prove Equation (2) and Equation (3) for $d = 0$.

By [Desfontaines et al., 2022, Lemma 1], we have that there exist $c_\ell$ and $c_h$ such that

$$\pi(c+1, \varepsilon, \delta) = \begin{cases} 0, & \text{if } c \leq 0 \\ e^\varepsilon \pi(c, \varepsilon, \delta) + \delta, & \text{if } 0 < c \leq c_\ell, \\ 1 - e^{-\varepsilon}(1 - \pi(c, \varepsilon, \delta) - \delta), & \text{if } c_\ell < c \leq c_h, \\ 1, & \text{if } c > c_h. \end{cases}$$

Moreover, the above implies $\pi(1, \varepsilon, \delta) = \delta$, $\pi(c_h, \varepsilon, \delta) \in [1 - \delta, 1)$.

We start by proving Equation (2). We show that

$$\forall m \geq 2, \qquad P_0(m) \leq e^\varepsilon P_1(m). \tag{4}$$

Since, in addition, $\Pr_{m \sim P_0}(m \leq 1) = P_0(1) = \pi(1, \varepsilon, \delta) - \pi(0, \varepsilon, \delta) = \delta$, Equation (2) holds.

To see Equation (4), when $0 \leq m - 2 \leq c_\ell$, we have

$$\begin{aligned} P_0(m) &= \pi(m, \varepsilon, \delta) - \pi(m - 1, \varepsilon, \delta) \\ &= \pi(m, \varepsilon, \delta) - (e^\varepsilon \pi(m - 2, \varepsilon, \delta) + \delta) \\ &\leq (e^\varepsilon \pi(m - 1, \varepsilon, \delta) + \delta) - (e^\varepsilon \pi(m - 2, \varepsilon, \delta) + \delta) \\ &\leq e^\varepsilon (\pi(m - 1, \varepsilon, \delta) - \pi(m - 2, \varepsilon, \delta)) \\ &= e^\varepsilon P_1(m), \end{aligned} \tag{5}$$

where Equation (5) is due to the $(\varepsilon, \delta)$-DP guarantee of $\pi$.

If $c_h \geq m - 2 > c_\ell$, we have

$$\begin{aligned} P_0(m) &= \pi(m, \varepsilon, \delta) - \pi(m - 1, \varepsilon, \delta) \\ &= \pi(m, \varepsilon, \delta) - (1 - e^{-\varepsilon}(1 - \pi(m - 2, \varepsilon, \delta) - \delta)) \\ &\leq (1 - e^{-\varepsilon}(1 - \pi(m - 1, \varepsilon, \delta) - \delta)) - (1 - e^{-\varepsilon}(1 - \pi(m - 2, \varepsilon, \delta) - \delta)) \\ &\leq e^{-\varepsilon}(\pi(m - 1, \varepsilon, \delta) - \pi(m - 2, \varepsilon, \delta)) \\ &= e^{-\varepsilon} P_1(m), \end{aligned} \tag{6}$$

where Equation (6) follows since, by the $(\varepsilon, \delta)$-DP guarantee of $\pi$, we have $1 - \pi(m - 1, \varepsilon, \delta) \leq e^\varepsilon(1 - \pi(m - 1, \varepsilon, \delta)) + \delta$. For $m - 2 > c_h$, we have $P_0(m) = \pi(m, \varepsilon, \delta) - \pi(m - 1, \varepsilon, \delta) = 0$. Combining the three cases completes the proof of Equation (2).

To prove Equation (3), we similarly need to show that

$$\forall m \leq c_h + 1, \qquad P_1(m) \leq e^\varepsilon P_0(m), \tag{7}$$

and then since $\Pr_{m \sim P_1}(m \geq c_h + 2) = P_1(c_h + 2) = \pi(c_h + 1, \varepsilon, \delta) - \pi(c_h, \varepsilon, \delta) \leq 1 - (1 - \delta) = \delta$, Equation (3) will follow. The proof of Equation (7) follows the proof of Equation (4) and is omitted here. $\qquad \square$

## Acknowledgments and Disclosure of Funding

The authors thank Itai Dinur for helpful conversations about this work. Haim Kaplan and Uri Stemmer are partially supported by the Israel Science Foundation (grants 1156/23 and 1419/23) and the Blavatnik family foundation.

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

# Appendices

## A  Cap Estimation

Suppose that each user contributes an arbitrary number of items to the dataset. A simple way to compute a differentially private set union in this case is to fix a cap $k$, enforce the cap by random selection (that is, if a user contributes more than $k$ items, retain $k$ of them uniformly at random and discard the rest), and finally apply the simple budget splitting mechanism $M_{split}$.

But how should we choose $k$? If $k$ is too large, the budget will be overly subdivided; if $k$ is too small, then we will discard a large portion of the dataset. In either case we are likely to get poor utility. However, the ideal value of $k$ depends on the dataset, so we must consider the privacy implications of trying to find it. In this section we suggest a method to select $k$ that will not consume a substantial fraction of our privacy budget.

The idea is simple. For each value of $k$ we compute the expected utility of $M_{split}$ on the randomly capped dataset, and then we run the exponential mechanism Dwork et al. [2014] (or some private noisy maximum algorithm) to pick the value of $k$ that maximizes the expected utility. This will be successful if the sensitivity of the expected utility remains small even as $k$ grows. On the one hand, as $k$ increases, individual users can make more contributions to the capped dataset, which will tend to increase sensitivity. On the other hand, the per-item budget shrinks, reducing the effect of each contributed item. We will show that these effects cancel out such that the sensitivity can be bounded independently of $k$.

We first establish an upper bound on the increase in $\pi$ when the count of an item grows by one.

**Lemma A.1.** *For $c \geq 0$, $\pi(c+1; \varepsilon, \delta) - \pi(c; \varepsilon, \delta) \leq \frac{e^{\varepsilon}-1}{2} + \delta$.*

*Proof.* Recall the recursive definition of $\pi$ from Desfontaines et al. [2022]:

$$\pi(0; \varepsilon, \delta) = 0$$
$$\pi(c+1; \varepsilon, \delta) = \min\left\{ e^{\varepsilon}\pi(c; \varepsilon, \delta) + \delta, 1 - e^{-\varepsilon}(1 - \pi(c; \varepsilon, \delta) - \delta), 1 \right\} \tag{8}$$

We have

$$\pi(c+1; \varepsilon, \delta) - \pi(c; \varepsilon, \delta) \leq \min\left\{ e^{\varepsilon}\pi(c; \varepsilon, \delta) + \delta, 1 - e^{-\varepsilon}(1 - \pi(c; \varepsilon, \delta) - \delta) \right\} - \pi(c)$$
$$= \min\left\{ (e^{\varepsilon} - 1)\pi(c; \varepsilon, \delta) + \delta, 1 + (e^{-\varepsilon} - 1)\pi(c; \varepsilon, \delta) - e^{-\varepsilon}(1 - \delta) \right\}.$$

As a function of $\pi(c; \varepsilon, \delta)$, the left term in the minimum is increasing and the right term is decreasing. Therefore the minimum is bounded by the value of the two terms when they agree:

$$(e^{\varepsilon} - 1)\pi(c; \varepsilon, \delta) + \delta = 1 + (e^{-\varepsilon} - 1)\pi(c; \varepsilon, \delta) - e^{-\varepsilon}(1 - \delta), \tag{9}$$

which implies

$$\pi(c; \varepsilon, \delta) = \frac{1}{e^{\varepsilon} + 1}(1 - \delta). \tag{10}$$

Plugging this back into the left term, we obtain the bound

$$\pi(c+1; \varepsilon, \delta) - \pi(c; \varepsilon, \delta) \leq \frac{e^{\varepsilon} - 1}{e^{\varepsilon} + 1}(1 - \delta) + \delta \leq \frac{e^{\varepsilon} - 1}{2} + \delta. \tag{11}$$

$\square$

Let $U(D, k)$ denote the expected utility of $M_{split}$ on a dataset $D$ after users are restricted to $k$ items by uniform random selection. The following result shows that the sensitivity of $U(D, k)$ can be bounded independently of $k$.

**Lemma A.2.** *For for all neighboring datasets $D' \sim D$ we have $|U(D', k) - U(D, k)| \leq \frac{e^{\varepsilon}-1}{2} + \delta$.*

*Proof.* Let $D_k$ denote the randomly capped version of $D$ where the bound of $k$ items per user has been enforced using uniform random selection. We have

$$U(D, k) = \mathop{\mathbb{E}}_{D_k}\left[\Pi(D_k; \varepsilon/k, \delta/k)\right] = \mathop{\mathbb{E}}_{D_k}\left[\sum_{x \in \mathcal{X}} \pi(c(x, D_k); \varepsilon/k, \delta/k)\right]. \tag{12}$$

Assume without loss of generality that $D'$ contains a user that $D$ does not. Let $S_k$ denote the random set of items contributed by the new user in $D'_k$. Because the $k$ bound is enforced independently for each user, we can couple $D'_k$ and $D_k$ using the pair $(D_k, S_k)$. Then $U(D', k) - U(D, k)$ is equal to

$$
\mathbb{E}_{D'_k}\left[\sum_{x \in \mathcal{X}} \pi(c(x, D'_k); \varepsilon/k, \delta/k)\right] - \mathbb{E}_{D_k}\left[\sum_{x \in \mathcal{X}} \pi(c(x, D_k); \varepsilon/k, \delta/k)\right]
$$

$$
= \mathbb{E}_{D_k, S_k}\left[\sum_{x \in \mathcal{X}} \left(\pi(c(x, D_k) + \mathbb{I}(x \in S_k); \varepsilon/k, \delta/k) - \pi(c(x, D_k); \varepsilon/k, \delta/k)\right)\right]. \quad (13)
$$

Because $|S_k| \leq k$, at most $k$ of the terms in the sum are nonzero, and we can apply Lemma A.1 to conclude that

$$
U(D', k) - U(D, k) \leq k\left(\frac{e^{\varepsilon/k} - 1}{2} + \frac{\delta}{k}\right) \leq \frac{e^\varepsilon - 1}{2} + \delta. \quad (14)
$$

$\square$

If $\varepsilon$ is small, then $(e^\varepsilon - 1)/2 + \delta$ is about $\varepsilon/2 + \delta$, and therefore the sensitivity of $U(D, k)$ is also small. In this case we can run the exponential mechanism (or an approximate noisy maximum algorithm) to pick the cap $k$ that approximately maximizes $U(D, k)$. Concretely, let $\Delta = (e^\varepsilon - 1)/2 + \delta$ denote our upper bound on the sensitivity. Then the exponential mechanism with privacy parameter $\varepsilon'$ obtains a cap $k$ such that

$$
U(D, k) \geq \max_{k'} U(D, k') - \frac{2\Delta}{\varepsilon'} \log\left(\frac{|\mathcal{K}|}{\beta}\right)
$$

with probability $1 - \beta$, where $\mathcal{K}$ denotes the set of possible caps we are optimizing over. (The utility guarantees of approximate noisy maximum algorithms are similar.)

## B  Impossibility Results

This section contains complete proofs for the results stated in Section 2. Appendix B.1 contains proofs of Theorem 2.1 and Theorem 2.2, while Appendix B.2 contains the proof of Theorem 2.3

### B.1  Bounds from Tower Datasets

In this section we prove Theorem 2.1 and Theorem 2.2, which are our strongest upper bounds on mechanism utility ratios, but only demonstrate difficulty on datasets that are very difficult (i.e., on these datasets, $\Pi(D)$ is small compared to the size of the non-private union). The proofs are organized slightly differently compared to the sketches provided in the main body. In particular, rather than explicitly constructing the datset $D$ on which a mechanism $M$ has low utility, we construct a distribution over datasets and show that the mechanism's average utility on that distribution is low. This will imply that there exists a dataset in the support of the distribution for which the utility ratio of the mechanism is low, but the distributional approach simplifies a number of arguments when moving to the more involved construction used in the proof of Theorem 2.2.

Recall that our goal is to bound $\max_M \min_{D \in \mathcal{D}_k} \frac{\sum_S P_M(S|D)|S|}{\Pi(D, \varepsilon, \delta)}$, since minimum is smaller than the average, we have the following lemma.

**Lemma B.1.** *For any distribution $\mathcal{P}$ over datasets,*

$$
\max_M \min_D \frac{\sum_S P_M(S|D)|S|}{\Pi(D, \varepsilon, \delta)} \leq \max_M \mathbb{E}_{D \sim \mathcal{P}}\left[\frac{\sum_S P_M(S|D)|S|}{\Pi(D, \varepsilon, \delta)}\right].
$$

We will choose $\mathcal{P}$ to be a uniform distribution over a set of datasets with the same value of $\Pi(D, \varepsilon, \delta)$. To define the class of datasets, we need a few definitions.

**Definition B.1** (Tower dataset). *A dataset $D = \{B_i\}_{i=1}^h$ is a tower dataset of height $h$ if there is an ordering $o : [h] \to [h]$ such that $B_{o(i)} \subseteq B_{o(i+1)}$ for each $i \leq h - 1$. Furthermore, we call $\bar{b}(D) = (|B_{o(1)}|, |B_{o(2)}|, \dots, |B_{o(h)}|)$ the shape of the dataset $D$. We denote $b_i = |B_{o(i)}|$. We omit $D$ and denote the shape by $\bar{b} = (b_1, \dots, b_h)$ when appropriate.*

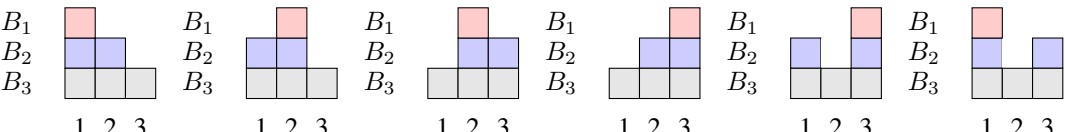

Figure 3: All possible tower datasets with three users with $b_1 = 1$, $b_2 = 2$, and $b_3 = 3$.

Notice that the datasets constructed in the proof sketch of Theorem 2.1 were tower datasets of shape $(1, k)$. In the rest of the section, unless otherwise stated all datasets are tower datasets and the elements are $\{1, 2, \ldots, k\}$. In Figure 3 we illustrate all possible tower datasets for a simple shape $\bar{b}$.

When $D$ is a tower dataset, the following result shows that $\Pi(D)$ depends only on the shape of $D$:

**Lemma B.2.** *For a tower dataset $D$ with shape $\bar{b}$ and height $h$, if $h \leq 1 + \left\lfloor \frac{1}{\varepsilon} \ln \left( \frac{e^\varepsilon + 2\delta - 1}{(e^\varepsilon + 1)\delta} \right) \right\rfloor$, then*

$$\Pi(D) = \delta \cdot \sum_{r=1}^{h} b_r e^{(h-r)\varepsilon}.$$

*Proof.* Note that there are $b_1$ elements that appear $h$ times and $b_2 - b_1$ elements that appear $h - 1$ times and so on. Furthermore, all the counts are smaller than the bound $c_\ell$ in Equation (1). Hence,

$$\Pi(D) = b_1 \pi(h) + \sum_{r=1}^{h-1} (b_{r+1} - b_r) \pi(h - r)$$

$$= \delta \left( b_1 \frac{e^{h\varepsilon} - 1}{e^\varepsilon - 1} + \sum_{r=1}^{h-1} (b_{r+1} - b_r) \frac{e^{(h-r)\varepsilon} - 1}{e^\varepsilon - 1} \right)$$

$$= \delta \sum_{r=1}^{h} b_r e^{(h-r)\varepsilon},$$

where the last equality follows by algebraic manipulation. $\square$

We now provide a complete proof for Theorem 2.1. Compared to the proof sketch in the main body of the paper, we adopt a proof technique that relies on the same key insights, but is slightly more aligned with the proof of Theorem 2.2 to help introduce the key ideas.

**Theorem 2.1** (Warm-up). *Let $k \geq 2$, $\varepsilon \geq 0$, $\delta \leq \frac{1}{e^\varepsilon + 2}$, and let $M$ be any $(\varepsilon, \delta)$-differentially private set union mechanism. Then there exists a dataset $D$ with contribution bound $k$ such that*

$$\frac{\mathbb{E}\big[|M(D)|\big]}{\Pi(D, \varepsilon, \delta)} \leq \frac{1}{k} + \frac{k}{e^\varepsilon + k}.$$

*In particular, for $\varepsilon = 2\ln(k)$ and $\delta \leq \frac{1}{k^2 + 2}$, we have $\frac{\mathbb{E}[|M(D)|]}{\Pi(D, \varepsilon, \delta)} \leq \frac{2}{k}$.*

*Proof.* We choose a uniform prior over all datasets of shape $(1, k)$ denoted by $\mathrm{unif}(1, k)$. Note that there are $k$ such datasets. For notational simplicity, let $D[i]$ denote the dataset containing two users, one contributing the singleton element $\{i\}$ and other contributing all $k$ items. Every dataset $D[i]$ has a single item that appears twice and $(k - 1)$ items that appear once. Therefore, $\Pi(D[i]) = \pi(2) + (k - 1)\pi(1)$. The bound on $\delta$ ensures that $2 \leq c_\ell$ from (1), which implies that $\pi(2) = e^\varepsilon \delta + \delta$. Together with the fact that $\pi(1) = \delta$ (regardless of parameters), we have that

$$\pi(D[i]) = \delta(e^\varepsilon + k), \tag{15}$$

for all $i \leq k$. Now consider the dataset $D[0]$ containing a single user $u$ whose bag of items $B(u) = \{1, 2, 3, \ldots, k\}$. This dataset is neighbor to all the datasets of shape $(1, k)$, hence for each such dataset $D[i]$

$$P(\{i\}|D[i]) \leq e^\varepsilon P(\{i\}|D[0]) + \delta, \tag{16}$$

and for all non-empty sets $S \neq \{i\}$,

$$\sum_{S \neq \{i\}} P(S|D[i]) \leq e^{\varepsilon} \left( \sum_{S \neq \{i\}} P(S|\tilde{D}[i]) \right) + \delta = \delta, \tag{17}$$

where $\tilde{D}[i]$ is a dataset with one user contributing only element $\{i\}$. Recall that $\mathrm{unif}(1,k)$ is the uniform distribution over all datasets of shape $(1,k)$ over elements $\{1,2,3,\ldots,k\}$.

Now fix any mechanism $M$ and for any item set $S \subset \mathcal{X}$ and dataset $D$, let $P(S \mid D)$ denote the probability that $M$ outputs $S$ when run on $D$. Then we have

$$\min_D \frac{\sum_S P(S \mid D) \cdot |S|}{\Pi(D, \varepsilon, \delta)} \leq \mathop{\mathbb{E}}_{D \sim \mathrm{unif}(1,k)} \left[ \frac{\sum_S P(S \mid D) \cdot |S|}{\Pi(D, \varepsilon, \delta)} \right]$$

$$\overset{(a)}{=} \frac{1}{\delta(e^{\varepsilon} + k)} \cdot \mathop{\mathbb{E}}_{D \sim \mathrm{unif}(1,k)} \left[ \sum_S P(S \mid D) \cdot |S| \right]$$

$$\overset{(b)}{=} \frac{1}{\delta(e^{\varepsilon} + k)} \cdot \left( \frac{1}{k} \sum_{i=1}^{k} \sum_S P(S \mid D[i]) \cdot |S| \right)$$

$$= \frac{1}{\delta(e^{\varepsilon} + k)} \cdot \frac{1}{k} \sum_{i=1}^{k} \left( P(\{i\} \mid D[i]) + \sum_{S \neq \{i\}} P(S \mid D[i]) \cdot |S| \right)$$

$$\overset{(c)}{\leq} \frac{1}{\delta(e^{\varepsilon} + k)} \cdot \frac{1}{k} \sum_{i=1}^{k} \left( P(\{i\} \mid D[i]) + \sum_{S \neq \{i\}} P_M(S \mid D[i]) \cdot k \right)$$

$$\overset{(d)}{\leq} \frac{1}{\delta(e^{\varepsilon} + k)} \cdot \frac{1}{k} \sum_{i=1}^{k} (P(\{i\} \mid D[0])e^{\varepsilon} + \delta + \delta k)$$

$$\overset{(e)}{\leq} \frac{1}{\delta(e^{\varepsilon} + k)} \cdot (\delta e^{\varepsilon}/k + \delta + \delta k)$$

$$= \frac{e^{\varepsilon} + k + k^2}{ke^{\varepsilon} + k^2}$$

$$= \frac{1}{k} + \frac{k}{e^{\varepsilon} + k},$$

where $(a)$ follows by Equation (15), $(b)$ follows by the definition of expectation, $(c)$ uses the fact that the size of each set is at most $k$. $(d)$ follows by Equations (16) (17). $(e)$ uses the fact that $\sum_{i=1}^{k} P(\{i\}|D[0]) \leq \delta$ as $D[0]$ has only one user. $\qquad \square$

Before moving on to the proof of Theorem 2.2, let us reexamine the techniques we used in this proof of Theorem 2.1.

1. We imposed a uniform prior over all datasets of shape $(1,k)$ to use Lemma B.1.

2. We divided the sets into two groups and for each particular group, we used differential privacy constraint w.r.t. a different neighboring dataset obtained by removing certain user from the dataset (e.g., Equations (16) (17)).

   (a) For nonempty sets $S \neq \{i\}$, we used DP constraint w.r.t. neighboring dataset $\tilde{D}[i]$ and used the fact that it assigns zero probability to all these sets.

   (b) For sets $\{i\}$, we used DP constraint w.r.t. neighboring dataset $D[0]$ and then summed over all such sets $\{i\}$ and finally used the fact that $D[0]$ has an empty neighboring dataset and hence the sum of probability it assigns to all such sets is at most $\delta$.

Our main upper bound follows a similar technique, but slightly more involved. We again impose a uniform prior on tower datasets of a certain shape with many users and we divide the sets into several groups and for each particular group, we use differential privacy constraint w.r.t. a different

neighboring dataset obtained by removing a certain user from the dataset. However, for the second step, we need to recursively remove several users to get to a neighboring dataset that assigns zero probability to that set. To formalize this intuition, we need the following two definitions. The first definition is that of a partial dataset, which is obtained by removing $i$ users from the dataset $D$ as follows:

**Definition B.2** (Partial dataset). *Let $r$ and $i$ be non-negative integers such that $r + i \leq h$. For a tower dataset $D = \{B_1, B_2, \ldots, B_h\}$ of height $h$, let*

$$D_i^r = \{B_1, B_2, B_r, B_{r+i+1}, B_{r+i+2}, \ldots, B_h\}.$$

*Note that height of $D_i^r$ is $h - i$.*

Notice that using this definition, we can start with $D$ and remove $B_{r+1}$ to get $D_1^r$ which is a neighbor of $D$, then we can continue and remove $B_{r+2}$ to get $D_2^r$, which is at distance 2 from $D$ and so on. In our proof, we note that we remove users in this particular order i.e., we start at a particular user $r$ and remove all users larger than $r$.

To get to a database for which we output $S$ with zero probability we have to remove all users that contain $S$. To this end, we define the rank of a set as follows.

**Definition B.3** (Rank of a set). *For a tower dataset $D$ of height $h$ and a set $S$, let $\mathrm{rank}(D, S)$ denote the number of users $i$ such that set $S$ is not contained in $B_i$ i.e.,*

$$\mathrm{rank}(D, S) = h - |\{i : S \subseteq B_i\}|.$$

*For example, if $D = \{(1), (1, 2), (1, 2, 3), (1, 2, 3, 4), (1, 2, 3, 4, 5)\}$ and $S = \{1, 3\}$, then $\mathrm{rank}(D, S) = 2$. For a dataset of height $h$, the rank ranges from $0$ to $h$.*

Note that because the user item sets are nested, we have that

$$\mathrm{rank}(D, S) = \max\{i \in [h] \mid S \not\subseteq B_i\},$$

since if $S$ is a subset of user $i$'s bag, it must also be a subset of $B_j$ for all $j \geq i$. In other words, $\mathrm{rank}(D, S)$ is the index of the last user in $D$ that does not contain every item in $S$. We have defined rank of a set w.r.t. a dataset as the number of users not containing the set, as opposed to number of users containing the set for notational simplification, since it ensures that the rank of $S$ does not change if we remove users containing $S$ in the particular order mentioned above.

To summarize, Definition B.2 allows us to discuss neighboring datasets where we remove user bags $B_{r+1}, B_{r+2}, \ldots, B_{r+i}$ and Definition B.3 allows us to discuss how many users do we need to remove to get a dataset for which there is zero probability to output $S$. We state the following set of properties for these partial datasets and rank, which will be useful in our proofs.

**Lemma B.3** (Properties of the rank and partial datasets).

1. *For any $r \leq h$,*

$$D_0^r = D. \tag{18}$$

2. *$D_i^r$ and $D_{i+1}^r$ are neighboring datasets.*

3. *If $\mathrm{rank}(D, S) = r$, then for any $i \leq h - r$*

$$\mathrm{rank}(D_i^r, S) = r. \tag{19}$$

4. *If $\mathrm{rank}(D, S) = r$, then $|S| \leq b_{r+1}$.*

5. *If $\mathrm{rank}(D, S) = r$, then*

$$P(S | D_{h-r}^r) = 0.$$

*Proof.* (1) follows from the fact that we have not removed any users. (2) uses the fact that $D_i^r$ and $D_{i+1}^r$ differ only in user $r + i + 1$. (3) follows from observing that if the rank of a set is $r$, then removing users $B_{r+i}$ for $i \geq 1$ does not change its rank. (4) uses the fact that $S \subseteq B_{r+1}$ and (5) follows by the fact that $D_{h-r}^r$ does not contain any sets that contain $S$ as the rank of $S$ is $r$. □

Suppose we have a uniform distribution over all datasets of a certain shape $\bar{b} = (b_1, b_2, \ldots, b_h)$. We draw a database from this distribution and then remove a user of particular size say $b_j$. Then we get a database of shape $\bar{b}_1^{j-1} = (b_1, \ldots, b_{j-1}, b_{j+1}, \ldots, b_h)$ with all datasets of these shape having the same probability to be the outcome. This is formalized by the following lemma.

**Lemma B.4.** *Let* $\text{unif}(\bar{b})$ *denote the uniform distribution over all datsets of shape* $\bar{b}$*. If* $D \sim \text{unif}(\bar{b})$ *then for any* $r, i$

$$D_i^r \sim \text{unif}(\bar{b}_i^r), \tag{20}$$

*where* $\bar{b}_i^r \triangleq (b_1, b_2, b_r, b_{r+i+1}, b_{r+i+2}, \ldots, b_h)$.

*Proof.* To simplify boundary cases, we let $B_0 = \emptyset$ and $B_{h+1} = \mathcal{X}$. We will argue that when we sample $D \sim \text{unif}(\bar{b})$ and obtain $D'$ by removing a single user from $D$, say user $B_j$, then $D'$ is a sample from $\text{unif}(\bar{b}_1^{j-1})$ where $\bar{b}_1^{j-1} = (b_1, \ldots, b_{j-1}, b_{j+1}, \ldots, b_h)$. Then the claim about removing several users follows by induction: We get $D_i^r$ by removing $i$ users from $D$, and each removal results in a uniform distribution over tower datasets of the relevant shape.

We now turn to proving the claim for removing a single user. Let $D \sim \text{unif}(\bar{b})$ and $D_{-j}$ denote the dataset obtained by removing user $j$. Similarly, for a concrete dataset $d$, let $d_{-j}$ denote the dataset obtained by removing user $j$ and for a shape $\bar{b}$, Recall that $\bar{b}_1^{j-1} = (b_1, \ldots, b_{j-1}, b_{j+1}, \ldots, b_h)$ denote the shape vector after removing the $j$th component. Let $\rho$ be the probability such that for every datset $d$ of shape $\bar{b}$, we have $\Pr(D = d) = \rho$. Now fix any dataset $d'$ of shape $\bar{b}_1^{j-1}$ and consider $\Pr(D_{-j} = d')$. By the Law of Total Probability, we have that

$$\Pr(D_{-j} = d') = \sum_{d \text{ of shape } \bar{b}} \Pr(D_{-j} = d' \mid D = d) \Pr(D = d)$$

$$= \rho \cdot \sum_{d \text{ of shape } \bar{b}} \mathbb{I}\{d_{-j} = d'\}.$$

In other words, the probability that $D_{-j} = d'$ is proportional to the number of datasets of shape $\bar{b}$ such that removing user $j$ from $d$ produces $d'$. We will count the number of such datasets. A datset $d$ has shape $\bar{b}$ and satisfies $d_{-j} = d'$ if and only if $d$ contains all the users of $d'$ together with a new user $B_j$ such that $|B_j| = b_j$ (so that the shape of $d$ is $\bar{b}$) and $B_{j-1} \subset B_j \subset B_{j+1}$ (so that $d$ is a tower dataset). The number of distinct choices for $B_j$ depends only on the shape $\bar{b}$: $B_j$ must contain $b_j$ distinct items selected from the $b_{j+1}$ items in $B_{j+1}$. And, it must include the $b_{j-1}$ items in $B_{j-1}$. So, the only freedom we have is to select $b_j - b_{j-1}$ items from $B_{j+1} \setminus B_{j-1}$ to include in $B_j$ in addition to $B_{j-1}$. There are $\binom{b_{j+1} - b_{j-1}}{b_j - b_{j-1}}$ ways to choose those items, giving

$$\Pr(D_{-j} = d') = \rho \cdot \binom{b_{j+1} - b_{j-1}}{b_j - b_{j-1}}.$$

Since this is true for all $d'$ of shape $\bar{b}_{-j}$, it follows that $D_{-j}$ is uniform over datasets of shape $\bar{b}_{-j}$. $\square$

Consider a set $S$ that has rank $r$ in dataset $D$. There is at least one item in $S$ that appears at most $h - r$ times in $D$, since otherwise the rank would be larger. This implies that the set $S$ can only be output by any private set union mechanism $M$ with probability at most $\pi(\text{rank}(D, S))$, since otherwise we would output at least one item with too high probability.

Consider a set $S$ for which we have to remove $h - r$ items to get a dataset which assigns zero probability to it. For items in this set, we expect the $\pi$ bound to be of the order of $\delta e^{(h-r-1)\varepsilon}$. However, we show that a single mechanism cannot assign a probability proportional to $\delta e^{(h-r-1)\varepsilon}$ for all such sets. To prove this, we use use the following contraction lemma.

**Lemma B.5.** *Let* $\text{unif}(\bar{b})$ *denote the uniform distribution over all datsets of shape* $\bar{b}$*. If* $D \sim \text{unif}(\bar{b})$*, then for any set* $S$ *and rank* $r \geq 0$ *and* $i \geq 0$ *such that* $r + i + 2 \leq h$*,*

$$\mathbb{E}[1_{\text{rank}(D_i^r, S) = r} | D_{i+1}^r] \leq \frac{b_{r+i+1}}{b_{r+i+2}} 1_{\text{rank}(D_{i+1}^r, S) = r} . \tag{21}$$

*In other words, for any instantiation of the dataset* $D_{i+1}^r$*, say* $d_{i+1}^r$*,*

$$\mathbb{E}[1_{\text{rank}(D_i^r, S) = r} | D_{i+1}^r = d_{i+1}^r] \leq \frac{b_{r+i+1}}{b_{r+i+2}} 1_{\text{rank}(d_{i+1}^r, S) = r} . \tag{22}$$

*Proof.* Fix a realization $d_{i+1}^r$ for the dataset $D_{i+1}^r$ and rewrite the left hand side of (22) as

$$\mathbb{E}[1_{\text{rank}(D_i^r,S)=r} \mid D_{i+1}^r = d_{i+1}^r] = \Pr(\text{rank}(D_i^r,S) = r \mid D_{i+1}^r = d_{i+1}^r).$$

We will consider three cases based on the rank of $S$ in $d_{i+1}^r$. First, suppose that $\text{rank}(d_{i+1}^r, S) < r$. Then we know that $S \subset B_r$ (since only at most the smallest $r-1$ users in $d_{i+1}^r$ do not contain $S$). But then from the tower dataset definition, we must have that $B_{r+i+1} \supset B_r \supset S$, which means we have added a new user to $d_{i+1}^r$ that also contains $S$, which does not change the rank of $S$ and we have $\text{rank}(D_i^r, S) < r$. In particular, both sides of (22) are zero and the inequality holds. Next, suppose that $\text{rank}(d_{i+1}^r, S) > r$. Then we know that $S$ is not contained in $B_{r+i+2}$, (since at most the $r$ smallest users in $d_{i+1}^r$ do not contain $S$). But again by the tower dataset definition, we have that $B_{r+i+1} \subset B_{r+i+2}$, which implies that $S \not\subset B_{r+i+1}$ and so the rank of $S$ in $D_i^r$ can only increase, so $\text{rank}(D_i^r, S) > r$. In particular, both sides of (22) are zero and the inequality holds. It remains to handle the case when $\text{rank}(d_{i+1}^r, S) = r$.

When $\text{rank}(d_{i+1}^r, S) = r$, the right hand side of (22) is equal to $\frac{b_{r+i+1}}{b_{r+i+2}}$, so we need to argue that this is an upper bound on the probability that $\text{rank}(D_i^r, S) = r$ conditioned on $D_{i+1}^r = d_{i+1}^r$. Observe that in this case, $\text{rank}(D_i^r) = r$ iff $S \subset B_{r+i+1}$, so we need to determine the conditional probability of this event given $D_{i+1}^r$ and $\text{rank}(S, D_{i+1}^r) = r$. From Lemma B.4 together with the fact that $D$ is uniform over tower datasets of shape $\bar{b}$, we have that $D_i^r$ is uniform over datasets of shape $\bar{b}_i^r$. After conditioning on $D_{i+1}^r = d_{i+1}^r$, the only randomness left is the draw of $u_{r+i+1}$'s bag. Say that an item bag $R \subset \mathcal{X}$ is *valid* for user $u_{r+i+1}$ if $|R| = b_{r+i+1}$ and $B_{r+i} \subset R \subset B_{r+i+2}$ so that $D_i^r$ is a tower dataset of shape $\bar{b}_i^r$ iff $B_{r+i+1}$ is valid. Conditioned on $D_{i+1}^r$, the item bag for user $u_{r+i+1}$ is drawn uniformly random from the set of valid bags. So, we can calculate the probability of the event $S \subset u_{r+i+1}$ by counting the fraction of valid bags that contain $S$.

First, let's count the total number of valid bags. We need to choose $\alpha = b_{r+i+1} - b_r$ items from the $\beta = b_{r+i+2} - b_r$ items in $B_{r+i+2}$ that are not already in $B_r$ to obtain $B_{r+i+1}$ of the right size that does not violate the tower constraints. There are $\binom{\beta}{\alpha} = \frac{\beta}{\alpha}\binom{\beta-1}{\alpha-1}$ ways to do that.

Next, we upper bound the number of valid choices that contain the set $S$. Let $x$ be any item in $S$ that is not present in $B_r$. Then every valid choice that contains $S$ must also contain $x$, so it is sufficient to upper bound the number of valid bags that contain the single item $x$. The number of ways to choose a subset of $B_{r+i+2}$ that contains $x$ is $\binom{\beta-1}{\alpha-1}$.

Taken together, we have shown that for any $d_{i+1}^r$ for which $\text{rank}(S, d_{i+1}^r) = r$, we have

$$\Pr(S \subset B_{r+i+1} \mid D_{i+1}^r = d_{i+1}^r) \leq \frac{\binom{\beta-1}{\alpha-1}}{\frac{\beta}{\alpha}\binom{\beta-1}{\alpha-1}} = \frac{\alpha}{\beta} = \frac{b_{r+i+1} - b_r}{b_{r+i+2} - b_r} \leq \frac{b_{r+i+1}}{b_{r+i+2}},$$

where the final inequality follows from the fact that if $x/y \leq 1$ then $x/y \leq (x+z)/(y+z)$ for any non-negative $x, y, z$.

$\square$

The next theorem gives an upper bound on the utility ratio for any mechanism $M$ by its utility ratio on tower datasets. In the subsequent results, we optimize over $\bar{b}$ to get the best bounds for a given $k, \varepsilon, \delta$.

**Lemma B.6.** *For a given $\varepsilon, \delta$, let $h_{\max} \triangleq 1 + \left\lfloor \frac{1}{\varepsilon} \ln\left(\frac{e^\varepsilon + 2\delta - 1}{(e^\varepsilon + 1)\delta}\right) \right\rfloor$,*

$$\max_M \min_D \frac{\sum_S P_M(S|D)|S|}{\Pi(D, \varepsilon, \delta)} \leq \min_{h \leq h_{\max}} \min_{\bar{b}:h(\bar{b})=h, b_h=k} \frac{\sum_{r=1}^h b_r \left(\sum_{s=r}^h \frac{b_r}{b_s} e^{(s-r)\varepsilon}\right)}{\sum_{r=1}^h b_r e^{(h-r)\varepsilon}}.$$

*Proof.* By Lemma B.1, for any distribution $\mathcal{P}$,

$$\max_M \min_D \frac{\sum_S P_M(S|D)|S|}{\Pi(D, \varepsilon, \delta)} \leq \max_M \mathbb{E}_{D\sim\mathcal{P}}\left[\frac{\sum_S P_M(S|D)|S|}{\Pi(D, \varepsilon, \delta)}\right].$$

Let $\mathcal{P}$ be the uniform distribution over all tower datasets of shape $\bar{b}$ with height $h \leq h_{\max}$ over the elements $\{1, 2, \ldots, k\}$. Since any dataset generated by this distribution has shape $\bar{b}$, we can compute their $\Pi$ value by Lemma B.2. Hence, combining it with above equation, we get

$$\max_M \min_D \frac{\sum_S P_M(S|D)|S|}{\Pi(D, \varepsilon, \delta)} \leq \frac{1}{\delta \sum_{r=1}^h b_r e^{(h-r)\varepsilon}} \cdot \max_M \mathbb{E}_{D \sim \mathrm{unif}(\bar{b})} \left[ \sum_S P_M(S|D)|S| \right]. \quad (23)$$

In the rest of the proof, we prove the following bound

$$\max_M \mathbb{E}_{D \sim \mathrm{unif}(\bar{b})} \left[ \sum_S P_M(S|D)|S| \right] \leq \delta \left( \sum_{r=1}^h b_r \left( \sum_{s=r}^h \frac{b_r}{b_s} e^{(s-r)\varepsilon} \right) \right) \quad (24)$$

combining the above two equations and taking the minimum over all shapes $\bar{b}$ such that $b_h = k$ and $h(\bar{b}) \leq h_{\max}$ yields the theorem. To prove Equation (24), we divide sets based on their rank as follows. For notational simplicity, we drop the subscript $M$.

$$\begin{aligned}
\mathbb{E}_{D \sim \mathrm{unif}(\bar{b})} \left[ \sum_S P(S|D)|S| \right] &= \mathbb{E}_{D \sim \mathrm{unif}(\bar{b})} \left[ \sum_S \sum_{r=0}^h 1_{\mathrm{rank}(D,S)=r} P(S|D)|S| \right] \\
&= \sum_{r=0}^h \mathbb{E}_{D \sim \mathrm{unif}(\bar{b})} \left[ \sum_S 1_{\mathrm{rank}(D,S)=r} P(S|D)|S| \right] \\
&= \sum_{r=0}^h \mathbb{E}_{D_0^r \sim \mathrm{unif}(\bar{b})} \left[ \sum_S 1_{\mathrm{rank}(D_0^r,S)=r} P(S|D_0^r)|S| \right] \\
&\overset{(a)}{=} \sum_{r=0}^{h-1} \mathbb{E}_{D_0^r \sim \mathrm{unif}(\bar{b})} \left[ \sum_S 1_{\mathrm{rank}(D_0^r,S)=r} P(S|D_0^r)|S| \right] \\
&\overset{(b)}{\leq} \sum_{r=0}^{h-1} \mathbb{E}_{D_0^r \sim \mathrm{unif}(\bar{b})} \left[ \sum_S 1_{\mathrm{rank}(D_0^r,S)=r} P(S|D_0^r) b_{r+1} \right] \\
&= \sum_{r=0}^{h-1} b_{r+1} \mathbb{E}_{D_0^r \sim \mathrm{unif}(\bar{b})} \left[ \sum_S 1_{\mathrm{rank}(D_0^r,S)=r} P(S|D_0^r) \right],
\end{aligned}$$

where $(a)$ follows by Lemma B.3 item (5) and $(b)$ follows by Lemma B.3 item (4). We are now going to bound $\mathbb{E}_{D_0^r \sim \mathrm{unif}(\bar{b})} \left[ \sum_S 1_{\mathrm{rank}(D_0^r,S)=r} P(S|D_0^r) \right]$ for a given value of $r$.

$$\begin{aligned}
\mathbb{E}_{D_0^r \sim \mathrm{unif}(\bar{b})} &\left[ \sum_S 1_{\mathrm{rank}(D_0^r,S)=r} P(S|D_0^r) \right] \\
&\leq \mathbb{E}_{D_0^r \sim \mathrm{unif}(\bar{b})} \left[ \delta + \sum_S 1_{\mathrm{rank}(D_0^r,S)=r} e^\varepsilon P(S|D_1^r) \right] \\
&= \delta + \sum_S e^\varepsilon \mathbb{E}_{D_0^r \sim \mathrm{unif}(\bar{b})} \left[ P(S|D_1^r) 1_{\mathrm{rank}(D_0^r,S)=r} \right] \\
&\overset{(a)}{=} \delta + \sum_S e^\varepsilon \mathbb{E}_{D_1^r \sim \mathrm{unif}(\bar{b})} \left[ P(S|D_1^r) \mathbb{E}_{D_0^r} \left[ 1_{\mathrm{rank}(D_0^r,S)=r} | D_1^r \right] \right] \\
&\leq \delta + \frac{b_{r+1}}{b_{r+2}} e^\varepsilon \sum_S \mathbb{E}_{D_1^r \sim \mathrm{unif}(\bar{b})} \left[ P(S|D_1^r) 1_{\mathrm{rank}(D_1^r,S)=r} \right], \quad (25)
\end{aligned}$$

where the first inequality follows by noting that $D_0^r$ and $D_1^r$ are neighboring datasets and applying differential privacy constraint to the event of outputting a set of rank $r$ and the last inequality follows from Lemma B.5. $(a)$ follows by law of total expectation and Lemma B.4.

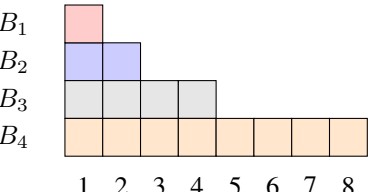

Figure 4: Example of the tower dataset in Theorem B.6 with $k = 8$, $h = 4$, $\varepsilon = 1$, and $\alpha = 1$.

Similarly, we can get $\forall i \leq h - r - 2$, we have

$$\mathbb{E}_{D_i^r \sim \text{unif}(\bar{b})} \left[ \sum_S 1_{\text{rank}(D_i^r, S) = r} P(S | D_i^r) \right]$$

$$\leq \delta + \frac{b_{r+i+1}}{b_{r+i+2}} e^\varepsilon \sum_S \mathbb{E}_{D_{i+1}^r \sim \text{unif}(\bar{b})} \left[ P(S | D_{i+1}^r) 1_{\text{rank}(D_{i+1}^r, S) = r} \right], \tag{26}$$

Substituting Equation (26) for $i = 1, \ldots, h - r - 2$ into Equation (25), we get that

$$\mathbb{E}_{D_0^r \sim \text{unif}(\bar{b})} \left[ \sum_S 1_{\text{rank}(D_0^r, S) = r} P(S | D_0^r) \right] \leq \delta \left( 1 + \frac{b_{r+1}}{b_{r+2}} e^\varepsilon + \frac{b_{r+1}}{b_{r+3}} e^{2\varepsilon} + \ldots \frac{b_{r+1}}{b_h} e^{(h-r-1)\varepsilon} \right).$$

Summing over all values of $r$ yields

$$\mathbb{E}_{D \sim \text{unif}(\bar{b})} \left[ \sum_S P(S | D) |S| \right] \leq \sum_{r=0}^{h-1} b_{r+1} \delta \left( 1 + \frac{b_{r+1}}{b_{r+2}} e^\varepsilon + \frac{b_{r+1}}{b_{r+3}} e^{2\varepsilon} + \ldots \frac{b_{r+1}}{b_h} e^{(h-r-1)\varepsilon} \right)$$

$$= \sum_{r=1}^{h} b_r \delta \left( 1 + \frac{b_r}{b_{r+1}} e^\varepsilon + \frac{b_r}{b_{r+2}} e^{2\varepsilon} + \ldots \frac{b_r}{b_h} e^{(h-r)\varepsilon} \right)$$

$$= \sum_{r=1}^{h} b_r \delta \left( \sum_{s=r}^{h} \frac{b_r}{b_s} e^{(s-r)\varepsilon} \right).$$

Combining the above equation with Equation (23) yields Equation (24) and hence the theorem.

$\square$

Next we move on to prove Theorem 2.2. We do this by finding a suitable value of $\bar{b}$ that provides the desirable upper bound.

Let $h = 1 + \lceil \frac{2}{\varepsilon} \log k \rceil$. For $r \geq 1$, let $b_r = \lceil e^{(r-1)\alpha\varepsilon} \rceil$ (see Figure 4 for an example). Let $\alpha$ be such that $b_h = k$. Note that

$$k = b_h = \lceil e^{(h-1)\alpha\varepsilon} \rceil \geq e^{(h-1)\alpha\varepsilon} > k - 1.$$

Hence, $\alpha \leq \frac{1}{\varepsilon(h-1)} \log k \leq \frac{1}{2}$. Note that the size of each set grows from $b_1 = 1$ to $b_h = k$ in the tower with $b_{i+1}/b_i \approx e^{\alpha\varepsilon}$. Furthermore,

$$1 + \left\lfloor \frac{1}{\varepsilon} \ln \left( \frac{e^\varepsilon + 2\delta - 1}{(e^\varepsilon + 1)\delta} \right) \right\rfloor \geq 1 + \left\lfloor \frac{1}{\varepsilon} \ln \left( \frac{e^\varepsilon - 1}{(e^\varepsilon + 1)\delta} \right) \right\rfloor \geq 1 + \left\lfloor \frac{2}{\varepsilon} \log k + 1 \right\rfloor \geq 1 + \left\lceil \frac{2}{\varepsilon} \log k \right\rceil \geq h,$$

where the first inequality uses the fact that $\delta \geq 0$ and the second inequality uses the upper bound on $\delta$ in the assumption of the theorem.

Using Lemma B.6, we get

$$\max_M \min_D \frac{\sum_S P_M(S | D) |S|}{\Pi(D, \varepsilon, \delta)} \leq \frac{\sum_{r=1}^{h} b_r \left( \sum_{s=r}^{h} \frac{b_r}{b_s} e^{(s-r)\varepsilon} \right)}{\sum_{r=1}^{h} b_r e^{(h-r)\varepsilon}}. \tag{27}$$

With $b_r = \lceil e^{(r-1)\alpha\varepsilon} \rceil$, and $x \leq \lceil x \rceil \leq 2x$ for $x \geq 1$, we get

$$\sum_{s=r}^{h} \frac{b_r}{b_s} e^{(s-r)\varepsilon} = \sum_{s=r}^{h} \frac{\lceil e^{(r-1)\alpha\varepsilon} \rceil}{\lceil e^{(s-1)\alpha\varepsilon} \rceil} e^{(s-r)\varepsilon}$$

$$\leq \sum_{s=r}^{h} \frac{2e^{(r-1)\alpha\varepsilon}}{e^{(s-1)\alpha\varepsilon}} e^{(s-r)\varepsilon}$$

$$= 2 \sum_{s=r}^{h} e^{(s-r)(1-\alpha)\varepsilon}$$

$$= 2 \frac{e^{(h-r+1)(1-\alpha)\varepsilon} - 1}{e^{(1-\alpha)\varepsilon} - 1}$$

$$\leq 2 \frac{e^{(h-r+1)(1-\alpha)\varepsilon}}{e^{(1-\alpha)\varepsilon} - 1}$$

Recall that $b_r \sum_{s=r}^{h} \frac{b_r}{b_s} e^{(s-r)\varepsilon}$ upper bounds the contribution of outputs of rank $r-1$, $r = 1, \ldots, h$ to the expected utility over $D \sim \mathrm{unif}(\bar{b})$. We see that $\sum_{s=r}^{h} \frac{b_r}{b_s} e^{(s-r)\varepsilon}$ decreases geometrically with $r$. But $b_r = \lceil e^{(r-1)\alpha\varepsilon} \rceil$ increases geometrically with $r$. So intuitively, since $\alpha$ is close to $1/2$, we get similar contribution from each rank. Plugging the above back into the ratio in Equation (27), we have

$$\max_M \min_D \frac{\sum_S P_M(S|D)|S|}{\Pi(D, \varepsilon, \delta)} \leq \frac{2}{e^{(1-\alpha)\varepsilon} - 1} \frac{\sum_{r=1}^{h} b_r e^{(h-r+1)(1-\alpha)\varepsilon}}{\sum_{r=1}^{h} b_r e^{(h-r)\varepsilon}}$$

$$\overset{(a)}{\leq} \frac{4}{e^{(1-\alpha)\varepsilon} - 1} \frac{\sum_{r=1}^{h} e^{(r-1)\alpha\varepsilon} e^{(h-r+1)(1-\alpha)\varepsilon}}{\sum_{r=1}^{h} e^{(r-1)\alpha\varepsilon} e^{(h-r)\varepsilon}}$$

$$= \frac{4}{e^{(1-\alpha)\varepsilon} - 1} \frac{e^{(h+1)(1-\alpha)\varepsilon} \sum_{r=1}^{h} e^{r\alpha\varepsilon} e^{-r(1-\alpha)\varepsilon}}{e^{h\varepsilon} \sum_{r=1}^{h} e^{r\alpha\varepsilon} e^{-r\varepsilon}}$$

$$= \frac{4}{e^{(1-\alpha)\varepsilon} - 1} \frac{e^{(1-\alpha)\varepsilon} \sum_{r=1}^{h} e^{r(2\alpha-1)\varepsilon}}{e^{\alpha h\varepsilon} \sum_{r=1}^{h} e^{r(\alpha-1)\varepsilon}}$$

$$\overset{(b)}{\leq} \frac{4e^{(1-\alpha)\varepsilon}}{e^{(1-\alpha)\varepsilon} - 1} \frac{he^{(2\alpha-1)\varepsilon}}{e^{\alpha h\varepsilon} \sum_{r=1}^{h} e^{r(\alpha-1)\varepsilon}}$$

$$= \frac{4he^{(2\alpha-1)\varepsilon}}{1 - e^{(\alpha-1)\varepsilon}} \frac{1 - e^{(\alpha-1)\varepsilon}}{e^{\alpha h\varepsilon} e^{(\alpha-1)\varepsilon} \left(1 - e^{h(\alpha-1)\varepsilon}\right)}$$

$$= \frac{4h}{e^{\alpha(h-1)\varepsilon} \left(1 - e^{h(\alpha-1)\varepsilon}\right)}$$

$$\overset{(c)}{\leq} \frac{12h}{k-1}$$

$$\leq \frac{12}{k-1} \left(1 + \frac{1}{\varepsilon} \log k\right),$$

where $(a)$ follows from $x \leq \lceil x \rceil \leq 2x$ since $b_r = \lceil e^{(r-1)\alpha\varepsilon} \rceil$, $(b)$ follows since $\alpha \leq 1/2$ implies that $e^{r(2\alpha-1)\varepsilon} \leq e^{(2\alpha-1)\varepsilon}$ for $r \geq 1$, and $(c)$ follows since $e^{(h-1)\alpha\varepsilon} \geq k-1$, and $e^{(h-1)\alpha\varepsilon} e^{h(\alpha-1)\varepsilon} \leq e^{\alpha\varepsilon} \leq \sqrt{k} \leq 2(k-1)/3$.

## B.2  Bounds from Fingerprinting Codes

In this section we prove Theorem 2.3.

**Theorem 2.3.** *Let $k \geq 2$, $n > 2 \cdot c_h$ (so that $\pi(n/2) = 1$), $\varepsilon \geq 1/n$, and $\delta < 1/(40e^\varepsilon n^{1/2} k^{1/4})$. Let $M$ be any $(\varepsilon, \delta)$-differentially private set union mechanism. Then there exists a dataset $D$ with contribution bound $k$ such that*

$$\frac{\mathbb{E}[|M(D)|]}{\Pi(D, \varepsilon, \delta)} = O\left(\left(\frac{n^2 \log(nk)}{k}\right)^{1/4}\right).$$

Note that Theorem 2.3 requires that we have at least $n = \Omega(\frac{1}{\varepsilon} \ln \frac{1}{\delta})$ users to ensure that $\pi(\frac{n}{2}; \varepsilon, \delta) = 1$, and the contribution bound $k$ must be relatively large compared to the number of users: $k = \Omega(n^2 \ln n)$.

**Remark B.7.** *Theorem 2.3 would hold even if we somewhat relax the definition of utility ratio. For example we can allow a small constant fraction ($\ll 1/2$) of the $\kappa\Pi(D, \varepsilon, \delta)$ items that we report to be mistakes. I.e. items that do not appear in the union. (Proving this will require a slight adaptation of Lemma B.8). We do not pursue this in this paper and leave investigating possible relaxations of the definition of utility ratio for future work.*

The key idea in the proof of Theorem 2.3 is a reduction showing that a private set union mechanism $A$ can be used to construct a mechanism for estimating the marginals of a matrix (see Definition B.4) whose performance can be bounded in terms of the utility ratio $u_k(A)$. Combined with an impossibility result for computing column-marginals based on robust fingerprinting codes, due to Steinke and Ullman [2015], we obtain our upper bound on the utility ratio.

**Definition B.4.** *Let $C \in \{0, 1\}^{n \times \ell}$ be a matrix and $A : \{0, 1\}^{n \times \ell} \to \{0, 1\}^{\ell}$ be a possibly randomized algorithm. We say that $A$ is $(\beta, \gamma)$-inaccurate for computing marginals of $C$ if the following holds with probability at least $\gamma$: Except on at most $\beta\ell$ coordinates $i \in [\ell]$, if the $i^{th}$ column of $C$ is all ones or zeros then $M(C)_i = 1$ or $M(C)_i = 0$, respectively. In other words, $M$ must identify the columns of $C$ that are all zeros or all ones, making a mistake on at most a $\beta$ fraction of the columns.*

Roughly speaking, our reduction views the matrix $C \in \{0, 1\}^{n \times \ell}$ as the description of a private set union instance where each column corresponds to an item and each row corresponds to a user. The $i^{th}$ row of $C$ is an indicator vector for the items in user $i$'s bag. The (non-private) union of the bags described above is exactly the set of columns for which outputting "1" is not a mistake. Given that we only start from a private set union mechanism that estimates the union imperfectly, the reduction outputs 1 for all columns in the approximate union, and outputs 0 or 1 with equal probability for each of the remaining columns. The following lemma connects between the utility ratio of the private set union mechanism and the $(\beta, \gamma)$-inaccuracy of the resulting matrix marginal mechanism.

**Lemma B.8.** *Let $M \in \text{UNION}_\ell(\varepsilon, \delta)$ be an $(\varepsilon, \delta)$-private set union mechanism with contribution bound $\ell$ and utility ratio $\kappa = u_\ell(M)$. Then for any $n$ large enough such that $\pi(n/2; \varepsilon, \delta) = 1$ and $\ell \in \mathbb{N}$, there exists an $(\varepsilon, \delta)$-differentially private matrix marginal mechanism $A : \{0, 1\}^{n \times \ell} \to \{0, 1\}^{\ell}$ that uses $M$ as a subroutine such that the following holds: For every matrix $C \in \{0, 1\}^{n \times \ell}$ such that at least half the columns of $C$ have at least $n/2$ ones, the mechanism $A$ is $(\beta, \gamma)$-inaccurate for computing the marginals of $C$ with $\beta = \frac{1}{2} - \frac{\kappa}{8} + \sqrt{\frac{\ln(40/\kappa)}{\ell} \cdot \left(\frac{1}{2} - \frac{\kappa}{8}\right)}$ and $\gamma = \frac{\kappa}{10}$.*

*Proof.* Fix any matrix $C \in \{0, 1\}^{n \times \ell}$ where at least half the columns have at least $n/2$ ones. We can transform a realization of $C$ into a private set union problem as follows: the item universe is $\mathcal{X} = \{1, \ldots, \ell\}$ and the dataset is $D$ has $n$ users, where user $i$ contributes bag $B_i = \{j \in \mathcal{X} \mid C_{ij} = 1\}$ is determined by interpreting row $i$ of $C$ as an indicator vector.

The matrix marginal mechanism $A$ works as follows: first, let $\hat{U} = M(D)$ be the private union output by $M$ when run on the dataset described by $C$. Then $A$ outputs a vector $y \in \{0, 1\}^{\ell}$ defined as follows: For each coordinate $j \in \hat{U}$, set $y_j = 1$. For the remaining coordinates $j \notin \hat{U}$, choose $y_j$ to be 0 or 1 with equal probability. Since $A$ post-processes the output of $M$, it is $(\varepsilon, \delta)$-differentially private, so it remains to bound the inaccuracy.

First, observe that the union $U^* = \bigcup_{i=1}^{n} B_i$ is exactly the set of columns that are not entirely zeros, which means that if $A$ outputs 1 for any column $i \in U^*$, it is not a mistake. Due to the subset requirement of private set union mechanisms, we are guaranteed that $\hat{U} \subset U^*$ and, since $A$ outputs 1 for the columns in $\hat{U}$, it never errs on columns in $\hat{U}$. On the other hand, for columns $j \notin \hat{U}$, $A$ flips a coin and will err with probability at most $1/2$, so we need to argue that $|\hat{U}|$ is large compared to $\ell$. By assumption, with probability one, at least half of the columns of $C$ contain at least $n/2$ ones, and $\pi(n/2; \varepsilon, \delta) = 1$. This implies that in the private set union instance $D$ derived from $C$, at least $\ell/2$ of the items appear at least $n/2$ times, and therefore $\Pi(D) = \sum_{j=1}^{\ell} \pi(\text{c}(j, D)) \geq \ell/2$.

Since $\kappa = u_\ell(M)$ is the utility ratio of the private set union mechanism $M$, we are guaranteed that $\mathbb{E}[|\hat{U}|] \geq \kappa\Pi(D) \geq \kappa\ell/2$. Note that since $|\hat{U}| \leq \ell$ with probability one, we also have

$\mathbb{E}[|\hat{U}|^2] \leq \ell \cdot \mathbb{E}[|\hat{U}|]$ Next we use Paley–Zygmund inequality to lower bound the probaiblity that $|\hat{U}| \geq \kappa\ell/4$.

**Lemma B.9** (Paley-Zygmund Inequality). *If $Z \geq 0$ is a random variable with finite variance, then for $0 \leq \theta \leq 1$, we have*

$$\Pr\left(Z \geq \theta\,\mathbb{E}[Z]\right) \geq (1-\theta)^2 \frac{\mathbb{E}[Z]^2}{\mathbb{E}[Z^2]}.$$

By Paley-Zygmund inequality, we have

$$\Pr\left(|\hat{U}| \geq \kappa\ell/4\right) \geq \frac{\kappa}{8}.$$

For a fixed $\hat{U}$, the number of errors $n_e$ the algorithm makes is a Binomial random variable with $\ell - |\hat{U}|$ trials and success probability $1/2$. We use Heoffding's inequality to prove the following concentration result on $n_e$.

**Lemma B.10** (Heoffding's Inequality). *Let $X, ..., X_n$ be independent random variables between 0 and 1. Consider the sum of these random variables, $S_n = \sum_{i=1}^{n} X_i$, we have*

$$\Pr(S_n - \mathbb{E}[S_n] \geq t) \leq \exp\left(-\frac{2t^2}{n}\right)$$

By Heoffding's inequality, we have

$$\Pr\left(n_e \geq \frac{\ell - |\hat{U}|}{2} + \sqrt{\frac{(\ell - |\hat{U}|)\log(40/\kappa)}{2}}\right) \leq \kappa/40.$$

Hence by union bound on the complements of the events $|\hat{U}| \geq \kappa\ell/4$ and $n_e \leq \frac{\ell - |\hat{U}|}{2} + \sqrt{\frac{(\ell - |\hat{U}|)\log(40/\kappa)}{2}}$, we upper bound their intersection as follows

$$\Pr\left(\frac{n_e}{\ell} \leq \frac{1}{2} - \frac{\kappa}{8} + \sqrt{\frac{\ln(40/\kappa)}{\ell} \cdot \left(\frac{1}{2} - \frac{\kappa}{8}\right)}\right) \geq \kappa/8 - \kappa/40 = \kappa/10,$$

completing the proof.

$\square$

The following hardness result for the problem of computing marginals follows from work on robust fingerprinting code Steinke and Ullman [2015]. We include its proof at the end of this section for completeness.

**Lemma B.11.** *There is no $(\varepsilon, \delta)$-DP algorithm with $\varepsilon = O(1), \delta < \gamma/(4e^\varepsilon n)$ which is $(\beta, \gamma)$-inaccurate for computing $\ell$-marginals on $n \times \ell$ matrices with at least $n/2$ ones in the majority of the columns for*

$$\beta \leq \frac{1}{2} - O\left(\left(\frac{n^2\left(\log(n) + 2\varepsilon + \log(1/\gamma)\right)}{\ell}\right)^{1/4}\right).$$

Equipped with this hardness result for privately computing marginals, we are ready to prove Theorem 2.3.

*Proof.* of Theorem 2.3. From Lemma B.8, it follows that an $(\varepsilon, \delta)$-private algorithm for set union with utility ratio $\kappa$ gives us an $(\varepsilon, \delta)$-private algorithm for $\ell$ marginals which is $(\beta, \gamma)$-inaccurate where

$$\beta = \frac{1}{2} - \frac{\kappa}{8} + \sqrt{\frac{\ln(40/\kappa)}{\ell} \cdot \left(\frac{1}{2} - \frac{\kappa}{8}\right)}$$

and $\gamma = \kappa/10$ on inputs matrices with $n$ rows, such that the majority of the columns contain at least $n/2$ ones, where $n$ is large enough such that $\pi(n/2) = 1$.

If $\delta \geq \gamma/(4e^{\varepsilon}n) = \kappa/(40e^{\varepsilon}n)$, by the assumption that $\delta < 1/(40e^{\varepsilon}n^{1/2}k^{1/4})$, we already have $\kappa = O\left(\left(\frac{n^2 \log(n/\kappa)}{k}\right)^{1/4}\right)$. When $\delta < \gamma/(4e^{\varepsilon}n)$, it follows from Lemma B.11 that

$$\frac{1}{2} - O\left(\left(\frac{n^2 \left(\log(n) + 2\varepsilon + \log(1/\gamma)\right)}{\ell}\right)^{1/4}\right) \leq \frac{1}{2} - \frac{\kappa}{8} + \sqrt{\frac{\ln(40/\kappa)}{\ell} \cdot \left(\frac{1}{2} - \frac{\kappa}{8}\right)}.$$

Rearranging the terms and substituting $\gamma = \kappa/10$, we get

$$\kappa = O\left(\left(\frac{n^2 \log(n/\kappa)}{\ell}\right)^{1/4}\right) + O(\sqrt{\frac{\log(1/\kappa)}{\ell}}) = O\left(\left(\frac{n^2 \log(n/\kappa)}{\ell}\right)^{1/4}\right).$$

This implies that $\kappa = O\left(\left(\frac{n^2 \log(n\ell)}{\ell}\right)^{1/4}\right)$. We conclude the proof by noting that the contribution bound $k \leq \ell$ in the construction. $\qquad\square$

Finally, we turn to the proof of Lemma B.11, which is a variation on the bound provided by Steinke and Ullman [2015].

*Proof.* of Lemma B.11

We use the robust fingerprinting codes from Steinke and Ullman [2015] defined as follows (Definition 2.20 in Steinke and Ullman [2015]).

**Definition B.5.** *A $n$-collusion resilient fingerprinting code of length $\ell$ for $m$ users robust to a $\beta$ fractions of errors, with failure probability $\zeta$, and false accusation fraction $\eta$, is a pair of random variables $C \in \{0,1\}^{m \times \ell}$ and $Trace : \{0,1\}^{\ell} \to 2^{[m]}$ such that the following holds. For all adversaries $Ad : \{0,1\}^{n \times \ell} \to \{0,1\}^{\ell}$ and $S \subset [m]$ with $|S| = n$ we have*

$$\mathcal{P}_{C,Trace,Ad}\Big[ \Big(\Big|\{1 \leq j \leq \ell \mid \nexists i \in [m], Ad(C_S)^j = c_i^j\}\Big| \leq \beta\ell\Big)$$
$$\wedge \left(Trace(Ad(C_S)) = \emptyset\right)\Big] \leq \zeta \qquad (28)$$

*where $C_S \in \{0,1\}^{n \times \ell}$ contains the rows of $C$ indexed by $S$, and the superscript $j$ is used to index the entries in vectors of length $\ell$, and*

$$\mathcal{P}_{C,Trace,Ad}\big[|Trace(Ad(C_S)) \cap ([m] \setminus S)| \geq \eta(m-n)\big] \leq \zeta \qquad (29)$$

Intuitively, Equation (28) says that if the adversary answers most columns correctly then Trace should accuse some users. More specifically, the first event in the intersection happens if the number of all ones columns $j$ on which the adversary says 0 ($Ad(C_S)^j = 0$) or vice versa is smaller than $\beta\ell$. The second event happens if Trace does not accuse a user. So we require that with probability $1 - \zeta$ either the adversary makes more than $\beta\ell$ mistakes or Trace accuses some users. (Note the stronger condition $i \notin [m]$ rather $i \notin [n]$ in Equation (28), this means that Trace is committed to answer on a larger set of responses of the adversary.) Equation (29) requires that Trace will not accuse more than $\eta$ fraction of the innocent users (that are not contained in $S$). Specifically, we require that with probability smaller than $\zeta$ Trace (wrongly) accuses more than $\eta$ fraction of the users who are not in $S$.

Steinke and Ullman [2015] constructed codes as specified in the following theorem (Theorem 2.21 in Steinke and Ullman [2015]).

**Theorem B.12.** *For every $1 \leq n \leq m$, $0 \leq \beta \leq 1/2$, and $0 \leq \eta \leq 1$, there is a $n$-collusion-resilient fingerprinting code of length $\ell$ for $m$ users robust to a $\beta$-fraction of errors with some fixed failure probability*

$$\zeta \leq \min\{\eta(m-n), 2^{-\Omega(\eta(m-n))}\} + \eta^{\Omega((1/2-\beta)n)}$$

*and false accusation fraction $\eta$ for*

$$\ell = O\left(\frac{n^2 \log(1/\eta)}{(1/2 - \beta)^4}\right) .$$

For lower bounds on DP algorithm we use such code with $m = n + 1$ users, and accusation fraction $\eta = O(1/n)$, since we want failure probability $\zeta = O(1/n)$. We call such a code *a $\beta$-robust, fingerprinting code for $n$ users with failure probability $\zeta$* (although the code is in fact for $n + 1$ users, $n$ of them colliding).

For our puprose, we need to modify the code of Theorem B.12 as follows.

**Theorem B.13.** *There exists a code with the same parameters as the code of Theorem B.12 such that the support of its matrices consists only of matrices in which the majority of the columns contain a majority of one and failure probability at most twice larger.*

*Proof.* We define a code $C'$ and $Trace'$ satisfying the requirements as follows. We generate $C$ and $Trace$ from the distribution of the code of Theorem B.12. Then if $C$ instantiates to a matrix in which less than half the columns contains more than half 1's we instantiate $C'$ to the complement of $C$ and instantiate $Trace'$ to flip its input before applying $Trace$. Otherwise we instantiate $C'$ and $Trace'$ to be equal to $C$ and $Trace$, respectively.

We claim that this new code $C'$ and $Trace'$ satisfies Conditions (28) and (29) of Definition B.5 with failure probability at most $2\zeta$ where $\zeta$ is the failure probability of original code $C, Trace$. We prove this by contradiction. Suppose there is an $Ad'$ that breaks $C', Trace'$. That is it either satisfies Condition (28) or Condition (29) with probability larger than $2\zeta$.

We construct an adversary $Ad$ for $C, Trace$ as follows. The adversary $Ad$ gets a set $S$ of $n$ row of $C$ and then flips a coin and with probability $1/2$ simply applies $Ad'$ to these $n$ rows and returns the same vector that $Ad'$ returns. Otherwise, it applies $Ad'$ to the complements of the $n$ rows that it got and returns the complement of what $Ad'$ returned. Let $S'$ be the set of rows to which we apply $Ad'$ ( $S' = S$ with probability 1/2 and otherwise $S' = \overline{S}$).

It follows from the definition of $C', Trace'$, that with probability $1/2$ $Ad$ applies $Ad'$ to a set $S'$ of $n$ row from the codebook $C'$ (the same distribution). Conditioned on this event, by our assumption, $Trace'$ would satisfy either Condition (28) or Condition (29) with probability at least $2\zeta$. When $C'$, $Trace'$ satisfy Condition (28) with respect to $Ad'$, then $C, Trace$ satisfy Condition (28) with respect to $Ad$. Similar claim holds for Condition (29).

This implies that $Ad$ fails with probability larger than $\zeta$ in its attack on $C, Trace$, which is a contradiction. $\qquad\square$

The existence of fingerprinting codes rules out DP algorithms which accurately compute marginals, described in the theorem below.

**Theorem B.14.** *If a $\beta$-robust, fingerprinting code of length $\ell$ for $n$ users with failure probability $\zeta$ exists, then for any $\varepsilon$ and $\delta$ such that*

$$e^{2\varepsilon}\zeta + e^{\varepsilon}\delta + \delta < \frac{\gamma - \zeta}{n}$$

*there is no $(\varepsilon, \delta)$-private, $(\beta, \gamma)$-inaccurate algorithm for computing the marginals of $\ell$-attributes of $n$-users. Furthermore, such a DP algorithm does not exists even if we require that it is $\beta$-inaccurate only on $n \times \ell$ matrices in which the majority of the columns contain more than $n/2$ ones.*

*Proof.* Let $M$ be a $(\varepsilon, \delta)$-private $\beta$-inaccurate algorithm for computing $\ell$ marginals of $n$ users. Define $Ad(C_S) := M(C_S)$ to be an adversary for the fingerprinting codes in Theorem B.13 that computes the marginals of the codewords of the users in $S$ (this is an $n \times \ell$ matrix) and responds with the result.

Consider the set of users $S = [n + 1] \setminus \{1\}$. Since $Ad(C_S)$ is correct on $(1 - \beta)\ell$ marginals with probability $\gamma$, and the complement of the event in Equation (28) holds with probability $1 - \zeta$, it follows that

$$\mathcal{P}_{C, Trace, Ad}\big[(Trace(Ad(C_S)) \neq \emptyset)\big] \geq \gamma - \zeta \,.$$

It follows that there exists some $i^* \in [n + 1] \setminus \{1\}$ such that

$$\mathcal{P}_{C, Trace, Ad}\big[(i^* \in Trace(Ad(C_S)))\big] \geq \frac{\gamma - \zeta}{n} \,.$$

Opening this up using the total probability formula, we get that

$$\sum_{B\in\{0,1\}^{m\times\ell}} \mathcal{P}_{Trace,Ad}\big[(i^* \in Trace(Ad(B_S)))\big] \cdot \mathcal{P}_C\big[B := C\big] \geq \frac{\gamma - \zeta}{n} \ . \tag{30}$$

Now consider the set of users $S' = [n+1] \setminus \{i^*\}$. By Equation (29) we get that

$$\mathcal{P}_{C,Trace,Ad}\big[(i^* \in Trace(Ad(C_{S'})))\big] \leq \zeta \ .$$

and again by the total probability formula we can write this as follows

$$\sum_{B\in\{0,1\}^{m\times\ell}} \mathcal{P}_{Trace,Ad}\big[(i^* \in Trace(Ad(B_{S'})))\big] \cdot \mathcal{P}_C\big[B := C\big] \leq \zeta \ . \tag{31}$$

Note that (1) $Trace(Ad()) = Trace(M())$ is $(\varepsilon, \delta)$-DP (Trace postprocess the output of the adversary, $Ad$, which is private), and (2) $|S \triangle S'| = 2$, so for every $B$ in the summations in Equations (30) and (31), the databases $B_S$ and $B_{S'}$ differ by two users. It follows that for every such pair $B_S, B_{S'}$

$$\mathcal{P}_{Trace,Ad}\big[(i^* \in Trace(Ad(B_S)))\big] \leq e^\varepsilon \left(e^\varepsilon \mathcal{P}_{Trace,Ad}\big[(i^* \in Trace(Ad(B_{S'})))\big] + \delta\right) + \delta \tag{32}$$

We use Equation (32) and (31) to upper bound the left hand side of Equation (30) by $e^\varepsilon(e^\varepsilon \zeta + \delta) + \delta$ so we get that

$$e^\varepsilon(e^\varepsilon \zeta + \delta) + \delta \geq \frac{\gamma - \zeta}{n} \ .$$

If $\varepsilon$ and $\delta$ do not satisfy this condition then we get a contradiction so $M$ cannot exist and the lemma follows. $\qquad\square$

Taking $\zeta = \frac{\gamma}{4ne^{2\varepsilon}}$, $\eta = 2\zeta = \frac{\gamma}{2ne^{2\varepsilon}}$, and $m = n+1$ in Theorem B.13, we have that there exist robust fingerprinting codes with code length

$$\ell = O\left(\frac{n^2 \log(1/\eta)}{(1/2 - \beta)^4}\right) = O\left(\frac{n^2 \left(\log(n) + 2\varepsilon + \log(1/\gamma)\right)}{(1/2 - \beta)^4}\right) \ .$$

When $\delta < \gamma/(4e^\varepsilon n)$, we have $e^\varepsilon(e^\varepsilon \zeta + \delta) + \delta < \frac{\gamma - \zeta}{n}$, and Lemma B.11 then follows from Theorem B.14. $\qquad\square$

## C  Algorithms with utility guarantees – missing proofs

We prove Lemma 3.1.

**Lemma 3.1.** *Let $M_{split}(D; \varepsilon, \delta, k)$ be the mechanism that works as follows: for each item $x \in \mathcal{X}$, include $x$ in the output with probability $\pi(c(x, D); \varepsilon/k, \delta/k)$. Then $M_{split}$ is a $(\varepsilon, \delta)$-differentially private set union mechanism when users contribute at most $k$ items.*

*Proof.* Since $\pi(0; \varepsilon/k, \delta/k) = 0$, items that do not appear in the input dataset are output with probability 0. It remains to check that $M_{split}$ is $(\varepsilon, \delta)$-differentially private.

The key idea is to think of $M_{split}$ as the composition of $|\mathcal{X}|$ simpler mechanisms, one for each item $x \in \mathcal{X}$. In particular, for each item $x \in \mathcal{X}$, let $M_x : \mathcal{D} \to \{0, 1\}$ be the mechanism that outputs 1 with probability $\pi(c(x, D), \varepsilon/k, \delta/k)$ and 0 otherwise. Then $M_{split}$ post-processes the output of the mechanisms $M_x$ by including the items whose mechanisms output 1 when run on $D$. Therefore, it is sufficient to prove that the composition of the mechanisms $(M_x)_{x\in\mathcal{X}}$ is $(\varepsilon, \delta)$-differentially private.

Fix any pair of neighboring datasets $D$ and $D'$. Since each user contributes at most $k$ items, we know that for all but at most $k$ of the items $x$, the mechanism $M_x$ has exactly the same output distribution on $D$ and $D'$, since $c(x, D) = c(x, D')$. Intuitively, it follows that when we apply a composition theorem to the collection $(M_x)_{x\in\mathcal{X}}$ that we only need to "pay" for $k$ of the mechanisms (instead of all $|\mathcal{X}|$). This is formalized in Lemma C.1. Finally, prior work establishes that, due to the definition of $\pi$, each mechanism $M_x$ is $(\varepsilon/k, \delta/k)$-differentially private. It follows that $M_{split}$ is $(\varepsilon, \delta)$-differentially private, as required. $\qquad\square$

The following easy lemma is needed in the proof of Lemma 3.1:

**Lemma C.1.** *Let $M_1 : \mathcal{D} \to \mathcal{Y}_1, \ldots, M_n : \mathcal{D} \to \mathcal{Y}_n$ be a collection of $(\varepsilon, \delta)$-differentially private mechanisms with each $\mathcal{Y}_i$ being finite. Say that $M_1, \ldots, M_n$ are $k$-aligned if for every pair of neighboring datasets $D$ and $D'$ we have that $M_i(D)$ has the same distribution as $M_i(D')$ except for at most $k$ indices $i \in [n]$. Then the composite mechanism $M(D) = \big(M_1(D), \ldots, M_n(D)\big)$ is $(k\varepsilon, k\delta)$-DP.*

*Proof.* Fix a pair of datasets $D$ and $D'$ and suppose without loss of generality that the mechanisms are numbered so that $M_i(D)$ has the same distribution as $M_i(D')$ for all $i > k$ (i.e., all of the mechanisms with different distributions are in the first $k$). Let $B = \mathcal{Y}_1 \times \ldots \times \mathcal{Y}_k$ and $A = \mathcal{Y}_{k+1} \times \ldots \times \mathcal{Y}_n$ and let $\Pi_A$ and $\Pi_B$ denote the projections from $\mathcal{Y}_1 \times \ldots \times \mathcal{Y}_n$ to $A$ and $B$, respectively.

For any output set $\mathcal{O} \subset \mathcal{Y}_1 \times \ldots \times \mathcal{Y}_n$ of the composite mechanism and any value $a \in A$ let $\mathcal{O}_{A=a}$ denote the set $\{(b, a) \mid b \in B \text{ and } (b, a) \in \mathcal{O}\}$, which is the "slice" of $\mathcal{O}$ where the components in $A$ are equal to $a$. Then we have

$$
\begin{aligned}
\Pr(M(D) \in \mathcal{O}) &= \sum_{a \in A} \Pr\big(M(D) \in \mathcal{O} \mid \Pi_A(M(D)) = a\big) \cdot \Pr\big(\Pi_A(M(D)) = a\big) \\
&= \sum_{a \in A} \Pr\big(\Pi_B(M(D)) \in \mathcal{O}_{A=a}\big) \cdot \Pr\big(\Pi_A(M(D)) = a\big) \\
&\leq \sum_{a \in A} \big(e^{k\varepsilon} \Pr\big(\Pi_B(M(D')) \in \mathcal{O}_{A=a}\big) + k\delta\big) \cdot \Pr\big(\Pi_A(M(D')) = a\big) \\
&= \left( e^{\varepsilon k} \sum_{a \in A} \Pr(\Pi_B(M(D')) \in \mathcal{O}_{A=a}) \cdot \Pr(\Pi_A(M(D')) = a) \right) + k\delta \\
&= e^{\varepsilon k} \Pr(M(D') \in \mathcal{O}) + k\delta,
\end{aligned}
$$

where the inequality follows from the fact that $\Pi_B(M(D))$ is the composition of $k$ $(\varepsilon, \delta)$-mechanisms and, by assumption, $\Pi_A(M(D))$ has the same distribution as $\Pi_A(M(D'))$. $\qquad\square$

## C.1 An alternative bicriteria algorithm

In this section, we present an alternative construction of a bicriteria algorithm. This algorithm achieves weaker guarantees compared to the one presented in Section 3.2, but it leverages different ideas, some of which might be useful in other applications. Before describing this alternative algorithm, we introduce tools from prior work that are needed for the construction.

**Sparse vector with individual charging.** Recall the celebrated *sparse vector technique* introduced by Dwork et al. [2009]: Given a dataset $D$ and a threshold $t$, the goal is to privately identify the first query $q_i$ out of a sequence of queries whose value on the dataset $D$ is "significant", i.e., $q_i(D) \gtrsim t$. Following Dwork et al. [2009], this technique was extended by Bun et al. [2017] to allow for identifying the first query that whose value is "close" to the threshold $t$. More specifically, they introduced an algorithm called `BetweenThresholds` with the following properties: In each round $i = 1, 2, 3, \ldots$, when getting the next query $q_i$, algorithm `BetweenThresholds` guarantees that:

**1)** If $q_i(D) \ll t$ then the algorithm returns $\mathsf{L}$ and continues to the next round.

**2)** Else if $q_i(D) \gg t$ then the algorithm returns $\mathsf{H}$ and continues to the next round.

**3)** Else (i.e., $q_i(D) \approx t$) then the algorithm returns $\top$ and halts.

The surprising aspect here, as was first shown by Dwork et al. [2009], is that the privacy parameters of `BetweenThresholds` do not scale with the number of queries. That is, the algorithm maintains $(\varepsilon, \delta)$-DP no matter how many queries it received before halting. What if we need to identify more than 1 query, say the first $k$ queries whose value on $D$ is roughly $t$? The textbook approach for this would be to re-execute algorithm `BetweenThresholds` after every $\top$ answer, paying in composition.

Kaplan et al. [2021] showed that this can be significantly relaxed. They introduced a variant of the sparse vector technique called *individual charging*. Instead of halting after the first $\top$ answer, the algorithm *deletes* from $D$ all elements that "contributed" to this answer (at most $\approx t$ elements) and

continues. More specifically, the algorithm of Kaplan et al. [2021] processes the next query $q_i$ as `BetweenThresholds` with item (3) replaced by:

**3)** Else (i.e., $q_i(D) \approx t$) then the algorithm returns $\top$, deletes from $D$ all users $u$ such that $q_i(u) = 1$, and continues to the next round.[1] For the sake of this informal presentation, let us interpret $q_i(D) \approx t$ as $q_i(D) \in t \pm \Delta$ for some error parameter $\Delta$.

In other words, like with the standard `BetweenThresholds` algorithm, answers of type L and H are obtained "for free". But now a $\top$ answer does not halt the algorithm altogether; instead we only "pay" for it by deleting some of the items from the data (at most $\approx t$), and continue. For example, if there are $k$ queries with value $\approx t$ that involve *different* elements in $D$, then the algorithm of Kaplan et al. [2021] identifies *all* of these queries at the price of *one* execution of `BetweenThresholds` (instead of $k$ executions), thus avoiding the cost of composition. The algorithm of Kaplan et al. [2021] was later optimized by Cohen and Lyu [2023]; we will use their optimized version.

**Overview of our bicriteria approximation.** At a high level, the algorithm works as follows. Let $t = 2\Delta$, where $\Delta$ is as defined in Step **3)** above. With this choice of $t$, the algorithm only returns $\top$ on queries whose value is between $\Delta$ and $3\Delta$. Now, given a dataset $D$ containing items from a universe $\mathcal{X}$ we do the following: For every $x \in \mathcal{X}$, issue the counting query $q_x$ to the algorithm of Cohen and Lyu [2023] (this query simply counts the number of occurrences of $x$ in $D$). We then report all items $x$ for which we got an answer of H or $\top$. We show that this algorithm obtains the aforementioned approximation bound. This includes overcoming three main obstacles:

**1.** The algorithms of Kaplan et al. [2021], Cohen and Lyu [2023] do not operate with the optimal reporting probabilities $\pi$. That is, for an item $x$ with count $c(x)$, the probability that the algorithm returns $\top$ or H on the query $q_x$ is *strictly smaller* than $\pi(c(x); \varepsilon, \delta)$, since it operates using Laplace noise and thresholding. To overcome this, we bound the optimal reporting probabilities in terms of the Laplace reporting probabilities.

**2.** In the analysis we need to argue about the effect of elements that the algorithm deletes from $D$ at during runtime. We do this via a charging argument, showing that if $r$ elements are deleted throughout the execution, then the algorithm must have reported at least $\approx \frac{r}{k\Delta}$ elements.

**3.** The algorithm of [Cohen and Lyu, 2023] can return H or $\top$ even if $c(x) = 0$ (with very small probability). Therefore, we refrain from applying this algorithm to items with count zero. Furthermore, to compete with $\Pi(D, \varepsilon', \delta')$ we have to deterministically report items $x$ such that $\pi(c(x), \varepsilon', \delta') = 1$. Thus we refrain from applying the algorithm to these items as well. These exclusions require some care in our privacy analysis.

We now formally introduce the algorithm of [Kaplan et al., 2021, Cohen and Lyu, 2023].

---

**Algorithm 2** `BetweenThresholds` with Charging [Kaplan et al., 2021, Cohen and Lyu, 2023]

**Input:** Dataset $D$, "hit" budget $\tau > 0$, privacy parameter $\varepsilon > 0$, thresholds $t_\ell < t_r$.

    1. For every user $j \in D$ set $C_j = 0$

    2. For round $i = 1, 2, \ldots$ do:

        (a) Receive a counting query $f_i$

        (b) $\hat{f}_i \leftarrow f_i(D) + \text{Lap}(\frac{1}{\varepsilon})$

        (c) If $\hat{f} < t_\ell$ then return L. If $\hat{f} > t_r$ then return H. Otherwise:

            • For each $j \in D$ such that $f_i(B_j) = 1$ do:

                – $C_j \leftarrow C_j + 1$.

                – If $C_j = \tau$ then remove $B_j$ from $D$.

            • Return $\top$.

---

**Theorem C.2** (Cohen and Lyu [2023]). *Let $\varepsilon < \frac{1}{2}$ and let $t_\ell < t_r$ be such that $t_r - t_\ell \geq \frac{3}{\varepsilon}$. Algorithm 2 is $(6\varepsilon\tau, e^{-\tau/4})$-DP. In particular, for $\tau = 4\ln(\frac{1}{\delta})$ we get that Algorithm 2 is $(24\varepsilon\ln(\frac{1}{\delta}), \delta)$-DP.*

---

[1]Here we think of $q$ as a *counting query*, meaning that it is a predicate defined over the data domain $X$, and for a dataset $S \in X^*$ we define $q(S) = \sum_{x \in S} q(x)$.

We are now ready to formally present our bicriteria approximation algorithm.

---

**Algorithm 3** `Bicriteria`

---

**Notation:** Let $k$ denote the contribution bound, let $\mathcal{X}$ be a domain of items, and let $\Delta_{\mathcal{X},k} = \{B \subseteq \mathcal{X} : |B| \leq k\}$ denote the set of all possible bags of size at most $k$ from $\mathcal{X}$.

**Input:** Dataset $D \in \Delta_{\mathcal{X},k}^n$, privacy parameter $\varepsilon > 0$.

1. Instantiate `BetweenThresholds` (Algorithm 2) on $D$ with parameters $\tau = 4\ln(\frac{1}{\delta})$, privacy parameter $\hat{\varepsilon} = \frac{\varepsilon}{24\ln(1/\delta)}$, and thresholds $t_\ell = \frac{1}{\hat{\varepsilon}}\ln(\frac{1}{\delta})$ and $t_r = t_\ell + \frac{3}{\hat{\varepsilon}}$.

2. For each $x \in \mathcal{X}$ do:

   (a) If $c(x, D) = 0$ then do not report $x$ and proceed to the next iteration.
   (b) Else if $c(x, D) \geq t_r + \frac{1}{\hat{\varepsilon}}\ln(\frac{1}{\delta})$ then report $x$ and proceed to the next iteration.
   (c) Otherwise:
   - Feed the counting query $c(x, D)$ to Algorithm `BetweenThresholds` and obtain an answer $a$.
   - If $a = \mathsf{L}$ then do not report $x$ and proceed to the next iteration.
   - If $a \in \{\top, \mathsf{H}\}$ then report $x$ and proceed to the next iteration.

---

Similarly to Algorithm `Bicrit` (presented in Section 3.2), Algorithm 3 does not really need to explicitly traverse all $x \in \mathcal{X}$ as we can skip items to which no user contributes.

The next lemma captures the privacy guarantees of Algorithm 3. This mostly follows from the privacy guarantees of `BetweenThresholds`, as captured by Theorem C.2. The proof does require some care, however, in order to handle the fact that in Steps 2a and 2b of Algorithm 3 we make a deterministic decision (without issuing any query to algorithm `BetweenThresholds`). This is necessary in order to allow us to later relate the reporting probabilities of our algorithm with the optimal reporting probabilities.

**Lemma C.3.** *Algorithm* `Bicriteria` *is* $(\varepsilon, O(ke^\varepsilon \delta))$-*DP.*

*Proof.* Fix two neighboring datasets $D^0$ and $D^1 = D^0 \cup \{B\}$ for $B = \{x_1, x_2, \ldots, x_z\}$ where $z \leq k$. Our goal is to show that

$$\texttt{Bicriteria}(D^0) \approx \texttt{Bicriteria}(D^1)$$

For the sake of the analysis, consider a modified variant of Algorithm `Bicriteria`, denoted as $\texttt{Bicriteria}_{D^0, D^1}$ and defined as follows. Algorithm $\texttt{Bicriteria}_{D^0, D^1}$ is identical to `Bicriteria`, except that in iterations on items $x$ such that $x \in B$, Algorithm $\texttt{Bicriteria}_{D^0, D^1}$ does not perform Steps 2a and 2b. Note that Algorithm $\texttt{Bicriteria}_{D^0, D^1}$ "knows" the two datasets $D^0, D^1$, and it is being applied to one of them. Also note that we only modify iterations corresponding to items $x \in B$.

We first argue that

$$\texttt{Bicriteria}_{D^0, D^1}(D^0) \approx_{(\varepsilon, \delta)} \texttt{Bicriteria}_{D^0, D^1}(D^1).$$

This follows as the outcome of $\texttt{Bicriteria}_{D^0, D^1}(D^b)$ can be viewed as a post-processing of the outcome of $\texttt{BetweenThresholds}(D^b)$. To see this, we design an algorithm $\mathcal{A}$ that knows $D^0, D^1$, but not $D^b$, and after interacting with $\texttt{BetweenThresholds}(D^b)$ it generates an outcome that is distributed exactly as the outcome of $\texttt{BetweenThresholds}(D^b)$. Algorithm $\mathcal{A}$ attempts to perform as much of the computation of $\texttt{Bicriteria}_{D^0, D^1}(D^b)$ by itself, and only interacts with $\texttt{BetweenThresholds}(D^b)$ when necessary. Specifically, Algorithm $\mathcal{A}$ mimics the loop of Step 2, and behaves differently on iterations where $x \in B$ and where $x \notin B$: If $x \in B$ then $\mathcal{A}$ queries $\texttt{BetweenThresholds}(D^b)$ and proceeds according to Step 2c (recall that in such iterations we do not perform Steps 2a and 2b). If $x \notin B$ then $\mathcal{A}$ can perfectly simulate this iteration *without accessing the data holder* $\texttt{BetweenThresholds}(D^b)$. In both cases $\mathcal{A}$ maintains the counts $C_j$, and excludes rows $j$ from the computation once they have reached their cap of $\tau$. This shows that $\texttt{Bicriteria}_{D^0, D^1}(D^b)$ can be written as a post-processing of $\texttt{BetweenThresholds}(D^b)$.

Next, note that the outcome distributions of $\texttt{Bicriteria}(D^b)$ and $\texttt{Bicriteria}_{D^0,D^1}(D^b)$ are within statistical distance $k\delta e^{\hat{\varepsilon}}$. To see this, note that throughout the execution of $\texttt{Bicriteria}_{D^0,D^1}(D^b)$ there are at most $z \le k$ iterations of Step 2 in which $\texttt{Bicriteria}_{D^0,D^1}(D^b)$ issues a query to Algorithm $\texttt{BetweenThresholds}$ even though $\texttt{Bicriteria}(D^b)$ would not have queried it (because it makes a deterministic decision in Steps 2a or 2b, which $\texttt{Bicriteria}_{D^0,D^1}(D^b)$ skips over). Recall that whenever $\texttt{BetweenThresholds}$ is queried, it samples one RV from the Laplace distribution. Define the good event $E$ stating that during all of these (at most $z$) queries to $\texttt{BetweenThresholds}$, the Laplace noises that are sampled are bounded in absolute value by $\frac{1}{\hat{\varepsilon}} \ln(\frac{1}{\delta}) - 1$. By the properties of the Laplace distribution and by a union bound, this event occurs with probability at least $1 - k\delta e^{\hat{\varepsilon}}$. When this event happens, in all of these (at most $z$) iterations, the answer reported by $\texttt{Bicriteria}_{D^0,D^1}(D^b)$ is identical to the (deterministic) answer reported by $\texttt{Bicriteria}(D^b)$. Furthermore, none of these queries to $\texttt{BetweenThresholds}$ results in $\top$, and thus do not change the internal state of $\texttt{BetweenThresholds}$. This shows that $\texttt{Bicriteria}(D^b)$ and $\texttt{Bicriteria}_{D^0,D^1}(D^b)$ are within statistical distance $k\delta e^{\hat{\varepsilon}}$.

Overall,

$$\texttt{Bicriteria}(D^0) \approx_{(0,k\delta e^{\hat{\varepsilon}})} \texttt{Bicriteria}_{D^0,D^1}(D^0)$$
$$\approx_{(\varepsilon,\delta)} \texttt{Bicriteria}_{D^0,D^1}(D^1)$$
$$\approx_{(0,k\delta e^{\hat{\varepsilon}})} \texttt{Bicriteria}(D^1),$$

showing that $\texttt{Bicriteria}$ is $(\varepsilon, \delta(1 + ke^{\hat{\varepsilon}} + ke^{\varepsilon+\hat{\varepsilon}}))$-DP. $\qquad\square$

We now proceed with the utility analysis of Algorithm 3. As we mentioned, the first step here is to relate the optimal reporting probabilities with the reporting probabilities that arise from our algorithm, which we refer to as the *Laplace* reporting probabilities, defined as follows.

**Definition C.1.** *The reporting probability of the Laplace mechanism with parameters $\varepsilon, \delta$ are*

$$\pi_{\mathrm{Lap}}(n; \varepsilon, \delta) = \begin{cases} 0, & \text{if } n = 0 \\ \Pr\left[n + \mathrm{Lap}(\frac{1}{\varepsilon}) > \frac{1}{\varepsilon} \ln(\frac{1}{2\delta}) + 1\right], & \text{if } 1 \le n \le \left\lceil \frac{2}{\varepsilon} \ln(\frac{1}{2\delta}) \right\rceil + 1 \\ 1, & \text{else} \end{cases}$$

**Remark C.4.** *The parameters in Definition C.1 were chosen such that for $n = 1$ we have $\pi_{\mathrm{Lap}}(1; \varepsilon, \delta) = \delta$ and for $n' = \left\lceil \frac{2}{\varepsilon} \ln(\frac{1}{2\delta}) \right\rceil + 1$ we have $\pi_{\mathrm{Lap}}(n'; \varepsilon, \delta) = 1 - \delta$. The choice for $n'$ could be slightly improved by tuning it to $(1 - \delta)e^{-\varepsilon}$ rather than $(1 - \delta)$, which we did not do for simplicity.*

**Lemma C.5.** *Let $\varepsilon \le \frac{1}{4}$ and $\delta \le \frac{\varepsilon}{100}$. For every $n$ it holds that*

$$\pi_{\mathrm{Lap}}\left(n; 2\varepsilon, \frac{2\delta}{\varepsilon}\right) \ge \pi(n; \varepsilon, \delta).$$

*Proof.* We prove the lemma for the following definition of $\pi_{\mathrm{Lap}}$, which for our choice of $\varepsilon, \delta$ is point-wise smaller (for every $n$) compared to Definition C.1.

$$\pi_{\mathrm{Lap}}(n; \varepsilon, \delta) = \begin{cases} 0, & \text{if } n = 0 \\ \Pr\left[n + \mathrm{Lap}(\frac{1}{\varepsilon}) > \frac{1}{\varepsilon} \ln(\frac{1}{\delta})\right], & \text{if } 1 \le n \le \frac{2}{\varepsilon} \ln(\frac{1}{6\delta^2}) \\ 1, & \text{else} \end{cases}$$

The proof proceeds by case analysis based on the value of $n$. We begin with the case that $n \leq n_{\text{switch}}^{\text{Lap}} \triangleq \frac{1}{2\varepsilon} \ln(\frac{\varepsilon}{2\delta})$. In this case we have that

$$
\begin{aligned}
\pi_{\text{Lap}}\left(n; 2\varepsilon, \frac{2\delta}{\varepsilon}\right) &= \int_{-n+\frac{1}{2\varepsilon}\ln(\frac{\varepsilon}{2\delta})}^{\infty} \varepsilon \cdot \exp(-2\varepsilon x)\, dx \\
&= \frac{1}{2}\exp(2\varepsilon n - \ln(\frac{\varepsilon}{2\delta})) \\
&= \delta \cdot \frac{\exp(2\varepsilon n)}{\varepsilon} \\
&\geq \delta \cdot \frac{e^{\varepsilon n} - 1}{e^{\varepsilon} - 1} \geq \pi(n; \varepsilon, \delta).
\end{aligned}
$$

Let us now consider $n$ in the range $n \in [n_{\text{switch}}^{\text{Lap}}, n_{\text{switch}}^{\text{opt}}]$, where $n_{\text{switch}}^{\text{opt}} = 1 + \left\lfloor \frac{1}{\varepsilon}\ln(\frac{e^{\varepsilon}+2\delta-1}{(e^{\varepsilon}+1)\delta}) \right\rfloor$. In this range we have

$$
\begin{aligned}
\pi_{\text{Lap}}\left(n; 2\varepsilon, \frac{2\delta}{\varepsilon}\right) &= \frac{1}{2} + \int_{-n+\frac{1}{2\varepsilon}\ln(\frac{\varepsilon}{2\delta})}^{0} \varepsilon \cdot \exp(2\varepsilon x)\, dx \\
&= 1 - \frac{1}{2}\exp(-2\varepsilon n + \ln(\frac{\varepsilon}{2\delta})) \\
&= 1 - \frac{\varepsilon}{4\delta}\cdot\exp(-2\varepsilon n).
\end{aligned}
$$

Recall that $n_{\text{switch}}^{\text{opt}}$ is the first *crossover point* of $\pi$, and that for $n$ smaller than this we have that $\pi(n; \varepsilon, \delta) = \delta \cdot \frac{e^{\varepsilon n}-1}{e^{\varepsilon}-1}$. Thus, we need to show that for $n \in [n_{\text{switch}}^{\text{Lap}}, n_{\text{switch}}^{\text{opt}}]$ it holds that

$$
1 - \frac{\varepsilon}{4\delta}\cdot\exp(-2\varepsilon n) \geq \delta \cdot \frac{e^{\varepsilon n} - 1}{e^{\varepsilon} - 1}.
$$

Simplifying the above inequality, it suffices to show that

$$
f(n) \triangleq e^{2\varepsilon n}\left(\frac{\delta}{\varepsilon}e^{\varepsilon n} - 1\right) + \frac{\varepsilon}{4\delta} \leq 0.
$$

Note that the function $\frac{\delta}{\varepsilon}x^3 - x^2$ is increasing on $(-\infty, 0]$, decreasing on $(0, \frac{3\delta}{2\varepsilon})$, and increasing again on $(\frac{3\delta}{2\varepsilon}, \infty)$. So the maximum of $f(n)$ for $n \in [n_{\text{switch}}^{\text{Lap}}, n_{\text{switch}}^{\text{opt}}]$ must be taken at the endpoints. We calculate:

$$
f(n_{\text{switch}}^{\text{Lap}}) = \frac{\varepsilon}{2\delta}\cdot\left(\frac{\delta}{\varepsilon}\cdot\sqrt{\frac{\varepsilon}{2\delta}} - 1\right) + \frac{\varepsilon}{4\delta} = \frac{1}{2}\left(\sqrt{\frac{\varepsilon}{2\delta}} - \frac{\varepsilon}{2\delta}\right) \leq 0,
$$

where the last inequality holds whenever $\delta \leq \frac{\varepsilon}{2}$. Similarly,

$$
f(n_{\text{switch}}^{\text{opt}}) \leq e^{2\varepsilon}\left(\frac{e^{\varepsilon}+2\delta-1}{(e^{\varepsilon}+1)\delta}\right)^2 \cdot \underbrace{\left(\frac{\delta}{\varepsilon}\cdot e^{\varepsilon}\left(\frac{e^{\varepsilon}+2\delta-1}{(e^{\varepsilon}+1)\delta}\right) - 1\right)}_{(*)} + \frac{\varepsilon}{4\delta}
$$

It can be verified that the expression denoted by $(*)$ is at most $-\frac{1}{6}$ whenever $\varepsilon \leq \frac{1}{2}$ and $\delta < \frac{\varepsilon}{100}$, in which case

$$
\begin{aligned}
f(n_{\text{switch}}^{\text{opt}}) &\leq -\frac{e^{2\varepsilon}}{6}\left(\frac{e^{\varepsilon}+2\delta-1}{(e^{\varepsilon}+1)\delta}\right)^2 + \frac{\varepsilon}{4\delta} \\
&\leq -\frac{e^{2\varepsilon}}{6}\left(\frac{\varepsilon}{(e^{\varepsilon}+1)\delta}\right)^2 + \frac{\varepsilon}{4\delta} \\
&= \frac{\varepsilon}{\delta}\left(-\frac{e^{2\varepsilon}}{6(e^{\varepsilon}+1)^2}\cdot\frac{\varepsilon}{\delta} + \frac{1}{4}\right) \\
&\leq \frac{\varepsilon}{\delta}\left(-\frac{e^{2\varepsilon}}{6(e^{\varepsilon}+1)^2}\cdot 100 + \frac{1}{4}\right) \leq 0.
\end{aligned}
$$

We now proceed to study values for $n$ in the range $n \in [n_{\text{switch}}^{\text{opt}}, n_{\text{end}}^{\text{Lap}}]$, where $n_{\text{end}}^{\text{Lap}} \triangleq \frac{1}{\varepsilon} \ln(\frac{\varepsilon^2}{24\delta^2})$. As before, the reporting probability of Laplace in this range (and actually for every $n \geq n_{\text{switch}}^{\text{Lap}}$) is

$$\pi_{\text{Lap}}\left(n; 2\varepsilon, \frac{2\delta}{\varepsilon}\right) = 1 - \frac{\varepsilon}{4\delta} \cdot \exp(-2\varepsilon n).$$

As for $\pi$ in that range, first observe that the second *crossover point* of the optimal reporting probabilities, denoted $n_{\text{end}}^{\text{opt}}$, is larger than $n_{\text{end}}^{\text{Lap}}$. Indeed, when $\varepsilon \leq 1/2$,

$$n_{\text{end}}^{\text{opt}} \geq \frac{1}{\varepsilon} \ln\left(\frac{e^\varepsilon + 2\delta - 1}{(e^\varepsilon + 1)\delta}\right) + \frac{1}{\varepsilon} \ln\left(1 + \frac{e^\varepsilon - 1}{\delta} \cdot \left(1 - \frac{e^\varepsilon + \delta}{e^\varepsilon + 1}\right)\right) - 1$$

$$\geq \frac{1}{\varepsilon} \ln\left(\frac{\varepsilon}{3\delta}\right) + \frac{1}{\varepsilon} \ln\left(1 + \frac{\varepsilon}{\delta} \cdot \frac{1}{3}\right) - 1$$

$$\geq \frac{1}{\varepsilon} \ln\left(\frac{\varepsilon^2}{9\delta^2}\right) - 1$$

$$\geq \frac{1}{\varepsilon} \ln\left(\frac{\varepsilon^2}{24\delta^2}\right) = n_{\text{end}}^{\text{Lap}}.$$

Thus, for every $n \in [n_{\text{switch}}^{\text{opt}}, n_{\text{end}}^{\text{Lap}}]$ we have

$$\pi(n; \varepsilon, \delta) = \left(1 - e^{-n\varepsilon} \cdot e^{\varepsilon \cdot n_{\text{switch}}^{\text{opt}}}\right) \cdot \left(1 + \frac{\delta}{e^\varepsilon - 1}\right) + e^{-\varepsilon n} \cdot e^{\varepsilon \cdot n_{\text{switch}}^{\text{opt}}} \cdot \pi\left(n_{\text{switch}}^{\text{opt}}; \varepsilon, \delta\right)$$

$$\leq \left(1 - e^{-n\varepsilon} \cdot e^{\varepsilon \cdot n_{\text{switch}}^{\text{opt}}}\right) \cdot \left(1 + \frac{\delta}{e^\varepsilon - 1}\right) + e^{-\varepsilon n} \cdot e^{\varepsilon \cdot n_{\text{switch}}^{\text{opt}}} \cdot \frac{e^\varepsilon + \delta}{e^\varepsilon + 1}$$

$$= 1 + \frac{\delta}{e^\varepsilon - 1} - e^{-\varepsilon n} \cdot e^{\varepsilon \cdot n_{\text{switch}}^{\text{opt}}} \cdot \left(1 + \frac{\delta}{e^\varepsilon - 1} - \frac{e^\varepsilon + \delta}{e^\varepsilon + 1}\right)$$

$$= 1 + \frac{\delta}{e^\varepsilon - 1} - e^{-\varepsilon n} \cdot e^{\varepsilon \cdot n_{\text{switch}}^{\text{opt}}} \cdot \left(\frac{\delta}{e^\varepsilon - 1} + \frac{1 - \delta}{e^\varepsilon + 1}\right)$$

$$\leq 1 + \frac{\delta}{e^\varepsilon - 1} - e^{-\varepsilon n} \cdot \frac{e^\varepsilon + 2\delta - 1}{(e^\varepsilon + 1)\delta} \cdot \left(\frac{\delta}{e^\varepsilon - 1} + \frac{1 - \delta}{e^\varepsilon + 1}\right)$$

$$\leq 1 + \frac{\delta}{e^\varepsilon - 1} - e^{-\varepsilon n} \cdot \frac{e^\varepsilon + 2\delta - 1}{(e^\varepsilon + 1)\delta} \cdot \frac{1}{4}$$

$$\leq 1 + \frac{\delta}{e^\varepsilon - 1} - e^{-\varepsilon n} \cdot \frac{\varepsilon}{3\delta} \cdot \frac{1}{4}.$$

So it suffices to verify that

$$1 - \frac{\varepsilon}{4\delta} \cdot \exp(-2\varepsilon n) \geq 1 + \frac{\delta}{e^\varepsilon - 1} - e^{-\varepsilon n} \cdot \frac{\varepsilon}{3\delta} \cdot \frac{1}{4}$$

which holds whenever

$$-\frac{\varepsilon}{4\delta} \cdot \exp(-2\varepsilon n) \geq \frac{\delta}{\varepsilon} - e^{-\varepsilon n} \cdot \frac{\varepsilon}{3\delta} \cdot \frac{1}{4},$$

i.e., whenever,

$$e^{-\varepsilon n} \frac{\varepsilon}{4\delta}\left(\frac{1}{3} - e^{-\varepsilon n}\right) \geq \frac{\delta}{\varepsilon}.$$

Note that $\frac{1}{3} - e^{-\varepsilon n} \geq \frac{1}{6}$ in our current range. Thus, it suffices to verify that

$$e^{-\varepsilon n} \frac{\varepsilon}{24\delta} \geq \frac{\delta}{\varepsilon},$$

which holds for every $n \leq n_{\text{end}}^{\text{Lap}} = \frac{1}{\varepsilon} \ln(\frac{\varepsilon^2}{24\delta^2})$.

All in all, for all $n \leq n_{\text{end}}^{\text{Lap}}$ we have that

$$\pi_{\text{Lap}}\left(n; 2\varepsilon, \frac{2\delta}{\varepsilon}\right) \geq \pi(n; \varepsilon, \delta).$$

Finally, for $n > n_{\text{end}}^{\text{Lap}}$ we have that

$$\pi_{\text{Lap}}\left(n; 2\varepsilon, \frac{2\delta}{\varepsilon}\right) = 1 \geq \pi(n; \varepsilon, \delta).$$

□

We are now ready to analyze the utility of our bicriteria algorithm.

**Lemma C.6.** *The expected number or identified items in Algorithm* `Bicriteria` *is*

$$\Omega\left(\frac{\varepsilon}{k \cdot \ln(\frac{1}{\delta})}\right) \cdot \Pi\left(D, \Omega\left(\frac{\varepsilon}{\ln(1/\delta)}\right), \Omega\left(\frac{\varepsilon\delta}{\ln(1/\delta)}\right)\right)$$

*Proof.* Let us consider a variant of Algorithm `Bicriteria`, called `BicriteriaNoDel`, which is identical to `Bicriteria` except that in Step 1 we initialize Algorithm `BetweenThresholds` with parameter $\tau = \infty$. This means that rows are never deleted from $D$ during the execution of `BetweenThresholds` in `BicriteriaNoDel`.

The resulting algorithm `BicriteriaNoDel` is not DP (with satisfactory privacy parameters), but its utility is high, in the sense that it achieves the Laplace reporting probabilities. Specifically, for every dataset $D$ we have

$$\mathbb{E}[|\text{BicriteriaNoDel}(D)|] \geq \Pi_{\text{Lap}}\left(D, \Omega\left(\frac{\varepsilon}{\ln(1/\delta)}\right), \delta\right) \triangleq \sum_{x \in D} \pi_{\text{Lap}}\left(c(x); \Omega\left(\frac{\varepsilon}{\ln(1/\delta)}\right), \delta\right).$$

We begin by showing that the utility of `Bicriteria` is comparable to that of `BicriteriaNoDel`. To this end, let $r \in \mathbb{R}^{|\mathcal{X}|}$ denote the internal randomness of `Bicriteria` (or `BicriteriaNoDel`), which we represent as a vector consisting of $|\mathcal{X}|$ samples from the Laplace distribution. We write `Bicriteria`$_r$ or `BicriteriaNoDel`$_r$ to denote these algorithms after fixing $r$. We next argue that for every $r$ and $D$ it holds that

$$|\text{Bicriteria}_r(D)| \geq \frac{\tau}{2\Delta k} \cdot |\text{BicriteriaNoDel}_r(D)|,$$

where $\Delta = t_r + \frac{1}{\hat{\varepsilon}}\ln(\frac{1}{\delta}) = \Theta\left(\frac{1}{\varepsilon}\ln^2(\frac{1}{\delta})\right)$ is the maximal possible value for $c(x, D)$ with which we might issue a query to `BetweenThresholds` during an iteration of Step 2. To see this, let $A \subseteq \mathcal{X}$ denote the set of all points that are reported by `BicriteriaNoDel`$_r(D)$. During the execution of `Bicriteria`$_r(D)$, some of the items in $A$ might not get reported. Specifically, an item $a \in A$ might not get reported if previous iterations of the algorithm deleted rows from $D$ that involve $a$. Otherwise, $a$ would be reported just as in the execution of `BicriteriaNoDel`$_r(D)$.

**Definition C.2.** *We say that an item* $a \in A$ *is* compromised *if previous iterations of* `Bicriteria`$_r(D)$ *(before the iteration on* $a$*) deleted rows that involve the item* $a$.

Let $R \subseteq A$ denote the set of compromised items. Note that all items $a \in A \setminus R$ (which are not compromised) are reported by `Bicriteria`$_r(D)$. We now show at least $\frac{\tau}{k\Delta}|R|$ of the items in $A$ are reported by `Bicriteria`$_r(D)$.

To this end, recall that every row in the dataset contains at most $k$ items. Thus, in order to have $|R|$ compromised items we must delete at least $\frac{|R|}{k}$ rows from the dataset. In order for this to happen, the sum of the counters maintained by algorithm `BetweenThresholds` must be at least $\frac{|R|\tau}{k}$ (because we delete a row only once its counter reaches $\tau$). In order for this to be the happen, we must observe at least $\frac{|R|\tau}{k\Delta}$ iterations in which algorithm `BetweenThresholds` returns $\top$. (This is because the counters are only increased during iterations with a $\top$ answer, and at most $\Delta$ counters are increased). Finally recall that in iterations with a $\top$ answer we report the corresponding item. Thus, if there are $|R|$ compromised items, then Algorithm `Bicriteria`$_r(D)$ reports at least $\frac{|R|\tau}{k\Delta}$ items.

Now, if $|R| \geq \frac{|A|}{2}$ then the number of items reported by `Bicriteria`$_r(D)$ is at least $\frac{|R|\tau}{k\Delta} \geq \frac{|A|\tau}{2k\Delta}$, and otherwise this number is at least $\frac{|A|}{2}$ (as all non compromised items get reported). So in any case we have that

$$|\text{Bicriteria}_r(D)| \geq \frac{|A|\tau}{2\Delta k} = \frac{\tau}{2\Delta k} \cdot |\text{BicriteriaNoDel}_r(D)|.$$

Overall,

$$\mathbb{E}[|\texttt{Bicriteria}(D)|] \geq \frac{\tau}{2\Delta k} \cdot \mathbb{E}[|\texttt{BicriteriaNoDel}(D)|]$$

$$\geq \frac{\tau}{2\Delta k} \cdot \Pi_{\mathrm{Lap}}\left(D, \Omega\left(\frac{\varepsilon}{\ln(1/\delta)}\right), \delta\right)$$

$$\geq \frac{\tau}{2\Delta k} \cdot \Pi\left(D, \Omega\left(\frac{\varepsilon}{\ln(1/\delta)}\right), \Omega\left(\frac{\varepsilon\delta}{\ln(1/\delta)}\right)\right)$$

$$\geq \Omega\left(\frac{\varepsilon}{k \cdot \ln(\frac{1}{\delta})}\right) \cdot \Pi\left(D, \Omega\left(\frac{\varepsilon}{\ln(1/\delta)}\right), \Omega\left(\frac{\varepsilon\delta}{\ln(1/\delta)}\right)\right),$$

where the second-to-last inequality follows from Lemma C.5.

$\square$

## C.2 Extreme privacy regimes – omitted proofs

We start with the following straightforward lemma about $M_{\mathrm{all}}$.

**Lemma C.7.** *Let $M_{all}(D; \delta)$ be the mechanism that returns the union of items in $D$ with probability $\delta$ and the empty set otherwise. Then $M_{all}$ is a $(0, \delta)$-differentially private set union mechanism.*

*Proof.* It is clear that the output of $M_{\mathrm{all}}(D)$ is always a subset of the union of $D$, so it remains to check that $M_{\mathrm{all}}$ is $(0, \delta)$-differentially private. Fix any pair of neighboring datasets $D$ and $D'$ (in fact, the proof works for any pair of datasets, even if they are not neighbors) and let $U$ and $U'$ be their unions, respectively. Then for any output set $\mathcal{O}$, we have that

$$\Pr(M_{\mathrm{all}}(D) \in \mathcal{O}) = \mathbb{I}\{\emptyset \in \mathcal{O}\}(1 - \delta) + \mathbb{I}\{U \in \mathcal{O}\}\delta$$

$$= \mathbb{I}\{\emptyset \in \mathcal{O}\}(1 - \delta) + \mathbb{I}\{U' \in \mathcal{O}\}\delta + (\mathbb{I}\{U \in \mathcal{O}\} - \mathbb{I}\{U' \in \mathcal{O}\})\delta$$

$$\leq \Pr(M_{\mathrm{all}}(D') \in \mathcal{O}) + \delta,$$

as required.

$\square$

We now restate and prove Lemma 3.6.

**Lemma 3.6.** *Let $M_{large}(D; \varepsilon, \delta, k)$ be the following mechanism: let $\delta' = \delta - \min(\delta, 1/\varepsilon)$ and output the union of $M_{all}(D; \delta')$ and $M_{split}(D; \varepsilon, \delta - \delta', k)$. Then $M_{large}$ is an $(\varepsilon, \delta)$-differentially private set union mechanism. Furthermore, for any contribution bound $k$, dataset $D$ with contributions bounded by $k$, and privacy parameter $\delta$, we have that*

$$\lim_{\varepsilon \to \infty} \frac{\mathbb{E}[|M_{large}(D; \varepsilon, \delta, k)|]}{\Pi(D; \varepsilon, \delta)} = 1.$$

*Proof.* From basic composition together with the privacy guarantees from Lemma C.7 and Lemma 3.1, it follows that $M_{\mathrm{large}}$ is $(\varepsilon, \delta)$-DP. Next, since $M_{\mathrm{all}}$ and $M_{\mathrm{split}}$ both output subsets of their input dataset, the union of their outputs is also a subset of the input. It remains to prove the utility guarantee.

Let $D_1 = \{x \in \mathcal{X} \mid c(x, D) = 1\}$ be the set of items that appear in $D$ exactly once, and $D_{>1} = \{x \in \mathcal{X} \mid c(x, D) > 1\}$ be the set of items that appear in $D$ two or more times. Then we have that

$$\Pi(D; \varepsilon, \delta) = \sum_{x \in \mathcal{X}} \pi(c(x, D); \varepsilon, \delta)$$

$$\geq \sum_{x \in D_1} \pi(1; \varepsilon, \delta) + \sum_{x \in D_{>1}} \pi(2; \varepsilon, \delta)$$

$$= \delta \cdot |D_1| + \min(1, e^\varepsilon \delta + \delta, 1 - e^{-\varepsilon}(1 - 2\delta)) \cdot |D_{>1}|,$$

where the inequality follows from the fact that $\pi$ is non-decreasing and the last equality follows from the fact that $\pi(1; \varepsilon, \delta) = \delta$ and the recursive definition of $\pi(2; \varepsilon, \delta)$. (i.e. we have that $\pi(2) \leq e^\varepsilon \pi(1) + \delta$ and $(1 - \pi(2))e^\varepsilon + \delta \leq (1 - \pi(1))$) Taking the limit as $\varepsilon \to \infty$ we have that $\lim_{\varepsilon \to \infty} \Pi(D; \varepsilon, \delta) = \delta \cdot |D_1| + |D_{>1}|$.

Since we are interested in the utility of $M_{\text{large}}$ only when $\varepsilon \to \infty$ if suffices to determine the expected output size of $M_{\text{large}}$ when $\varepsilon > 1/\delta$. In this case, we have that $\delta' = \delta - 1/\varepsilon$ and $\delta - \delta' = 1/\varepsilon$, which gives:

$$\mathbb{E}[|M_{\text{large}}(D; \varepsilon, \delta, k)|] = \sum_{x \in \mathcal{X}} \Pr(x \in M_{\text{large}}(D; \varepsilon, \delta, k)).$$

For any item $x \in D_1$, we have that

$$\Pr(x \in M_{\text{large}}(D; \varepsilon, \delta, k)) \geq \Pr(x \in M_{\text{all}}(D; \delta - 1/\varepsilon)) = \delta - 1/\varepsilon.$$

Next, for any item $x \in D_{>1}$, we have that

$$\Pr(x \in M_{\text{large}}(D; \varepsilon, \delta, k)) \geq \Pr(x \in M_{\text{split}}(D; \varepsilon, 1/\varepsilon)) \geq \pi(2; \varepsilon/k, 1/\varepsilon)$$
$$= \min(1, e^{\varepsilon/k}/\varepsilon + 1/\varepsilon, e^{-\varepsilon/k}(1 - 2/\varepsilon)).$$

Putting it together, we have that

$$\mathbb{E}[|M_{\text{large}}(D; \varepsilon, \delta, k)|] \geq (\delta - 1/\varepsilon) \cdot |D_1| + \min(1, e^{\varepsilon/k}/\varepsilon + 1/\varepsilon, e^{-\varepsilon/k}(1 - 2/\varepsilon)) \cdot |D_{>1}|.$$

Taking the limit as $\varepsilon \to \infty$ gives that $\lim_{\varepsilon \to \infty} \mathbb{E}[|M_{\text{large}}(D; \varepsilon, \delta, k)|] = \delta \cdot |D_1| + |D_{>1}|$.

Since both limits exist and are equal, it follows that

$$\lim_{\varepsilon \to \infty} \frac{\mathbb{E}[|M_{\text{large}}(D; \varepsilon, \delta, k)|]}{\Pi(D; \varepsilon, \delta)} = 1,$$

as required. $\qquad\square$

