# OpenReview forum: "Private Set Union with Multiple Contributions"
_NeurIPS.cc/2025/Conference — NeurIPS 2025 spotlight_

### Official Review · Reviewer_mdhA · 2025-06-15

**Clarity:** 3
**Significance:** 4
**Originality:** 3
**Rating:** 5
**Confidence:** 2

**Summary:**

Assume each user owns a subset of at most $k$ items that belong to a large universe $\mathcal{X}$, the Differentially Private Set Union (I prefer calling it Differentially Private Partition Selection, see comments) problem is to compute the union of the items from all users while satisfying a reasonable definition of differential privacy (DP). Previous works show that one can construct an optimal $(\epsilon, \delta)$-DP algorithm when each user has exactly one item. However, designing an optimal $(\epsilon, \delta)$-DP algorithm when each user contributes up to $k$ items is an open problem.

This paper presents several key findings for this open problem. First, Instance-optimal algorithms are not generally possible when $k > 1$. Second, it is possible to have an appropriate algorithm for settings where some public prediction regarding the union is available. Finally, the paper presents several concrete algorithms under various public predictions, accompanied by detailed proofs of privacy and utility.

In summary, this paper provides important results for the Differentially Private Partition Selection problems, helping researchers better understand its privacy boundaries.

**Questions:**

I cannot find a topic for NIPS 2025 that meets the topic of this paper. Could the authors formally state that the topic of this paper is considered for NIPS 2025?

**Ethical Concerns:**

["NO or VERY MINOR ethics concerns only"]

**Final Justification:**

Thanks again for the authors feedback. I believe changing the name from Private Set Union to Differentially Private Set Union can reflect the specific privacy constraint that this paper targets. I suggest having a separate paragraph descrbing the definition of Private Set Union from the crypto field and explicitly state the difference.

**Limitations:**

Aside from the problem of meeting the topics of NIPS 2025, I identify that the definition of Private Set Union provided by the authors is not the one defined in the cryptography topic. Specifically, in cryptography, PSU considers several (typically two) parties each having a set, and the output of PSU is the union of sets without leveraging the intersections of these sets. The detailed definitions and constructions for two parties and multiple parties can be found in [1][2], respectively. I suggest changing the term used in this paper or having a separate description to clarify the differences between the two definitions.

I have to say that the reason I bid on this paper is that I am quite familiar with the cryptographic PSU, but not the one defined in this paper. I tried my best to read the paper based on my DP knowledge, and I believe this paper is well-written. I am looking forward to seeing feedback from other reviewers.

[1] Kolesnikov, V., Rosulek, M., Trieu, N., & Wang, X. Scalable private set union from symmetric-key techniques. ASIACRYPT 2019, pp. 636-666. Cham: Springer International Publishing.
[2] Liu, Xiang, and Ying Gao. Scalable multi-party private set union from multi-query secret-shared private membership test. ASIACRYPT 2023, pp. 237-271. Singapore: Springer Nature Singapore, 2023.

**Paper Formatting Concerns:**

No.

**Quality:**

3

**Strengths And Weaknesses:**

# Strengths

1. Important (impossible and possible) results for the Differentially Private Partition Selection problem.
2. Concrete algorithms under possible settings.
3. Detailed privacy and utility proofs.

# Weaknesses

1. I am wondering if the topic of this paper meets one of the topics of NIPS 2025.
2. Private Set Union has been a well-defined **cryptography** topic. I suggest changing the term "Private Set Union" to another one, e.g., Differentially Private Partition Selection.
3. No implementations and experiments are given in this paper. I believe this is a minor problem since this paper focuses mainly on theoretical results.

---

> ### Author Rebuttal · Authors · 2025-07-30
>
> Thank you for taking the time to review our work. We address your specific points below.
>
> **> I cannot find a topic for NIPS 2025 that meets the topic of this paper. Could the authors formally state that the topic of this paper is considered for NIPS 2025?**
>
> The relevant topic from the NeurIPS call for papers is "privacy" (under "Social and economic aspects of machine learning"). Differential privacy is a longstanding interest of the ML community and has consistently appeared at previous conferences. (For example, the same problem has been studied in Gopi et al. (2020), which was presented at ICML 2020.)
>
> **> Private Set Union has been a well-defined cryptography topic... I suggest changing the term used in this paper or having a separate description to clarify the differences between the two definitions**
>
> We agree and are open to changing the name. One option is to change the name to "Differentially private set union" to reflect the specific privacy constraint we are working with. This is also consistent with prior work such as Gopi et al. (2020), and Carvalho et al. (2022).

---

> > ### Comment · Reviewer_mdhA · 2025-08-01
> >
> > Thank authors for the response. I don't have futher questions. As my initial rating is positive, I'll keep it as it is.

---

### Official Review · Reviewer_wkMV · 2025-06-17

**Clarity:** 3
**Significance:** 3
**Originality:** 3
**Rating:** 5
**Confidence:** 4

**Summary:**

This paper studies the private set union problem in the context of differential privacy, where each user contributes a bag of up to $k$ items from a large universe. The goal is to compute a subset of the union of all user-contributed items while preserving $(\varepsilon, \delta)$-differential privacy. The challenge arises because reporting infrequent items can violate privacy guarantees. Unlike previous work, which focused on the case where $k = 1$, this paper generalizes the setting to $k > 1$ and provides the first theoretical utility bounds in this more general case.

The main reason the previous paper do not have a good theoretical analysis on the private set union is because that it is not trivial to formalize the utility. As mentioned by the authors, a simple example shows any $(\varepsilon,\delta)$-differentially private algorithm cannot output a subset with expected cardinality larger than $|\bigcup_{i} S_i| \cdot \delta$, so it does not make much sense to directly analyzing the size of the output. In this paper, the authors introduce the utility ratio, which normalizes the expected utility by a dataset-specific upper bound. This allows comparison of algorithms in terms of their worst-case normalized performance. Given this measurement, the authors give several impossibility results and matching algorithms.

**Questions:**

In this paper, for the union set case where $k = 1$, you refer to the sampling-based algorithm from the Cohen et al. paper. I believe there is another algorithm based on thresholding the histogram:

(i) Add Laplace noise with parameter $1/\varepsilon$ to the frequency of elements in $|\bigcup_{i} S_i|$;

(ii) Output all elements whose noisy frequency exceeds a threshold $T$, where $T$ is approximately $\log(1/\delta)/\epsilon$.

It should be easy to show that this algorithm is $(\epsilon,\delta)$-differentially private (see the proof of Lemma 5 in the paper “Releasing Search Queries and Clicks Privately,” WWW 2009 for details). Clearly, these two algorithms are not equivalent. However, I did some preliminary calculations by myself, and I believe the thresholding approach achieves asymptotically similar error bounds to the i.i.d. sampling algorithm in Cohen et al.

My motivation for comparing these two algorithms is that the thresholding method appears more intuitive. In particular, it provides a natural explanation for why $\pi$ is a three-piece function:

(i) the frequency almost surely falls below the threshold (corresponding to the case $c(x) \leq c_{\ell}$);

(ii) the frequency almost surely exceeds the threshold (corresponding to the case $c(x) > c_{h}$); and

(iii) the intermediate case.

So if they are indeed equivalent (in terms of the utility), then perhaps using the thresholding algorithms as the benchmark would be better? What do you think?

**Ethical Concerns:**

["NO or VERY MINOR ethics concerns only"]

**Final Justification:**

The authors properly respond my questions.

**Limitations:**

Yes.

**Paper Formatting Concerns:**

No.

**Quality:**

3

**Strengths And Weaknesses:**

This paper considers an important problem in differential privacy and presents several clearly stated theoretical results. In fact, to the best of my knowledge, it is the first paper to provide any form of theoretical utility bounds for private set union with $k > 1$. (The work by Gopi et al. analyzes only the privacy guarantees, while the Cohen et al. paper implicitly handles only the special case where $k = 1$.) Here, $k$ denotes the maximum cardinality of the input subsets. Moreover, the paper is well-written and easy to follow. The proof sketches included in the main text are especially helpful in conveying the core techniques and proof ideas.

In my view, the most intuitive way to define “utility” in the context of private set union is to aim to preserve all items $x$ with reasonably large frequency $c(x)$. The function $\pi(c(x))$ offers a strong intuition for how frequency impacts utility, making it natural to define a utility ratio for the general case of $k > 1$. This, in turn, provides a meaningful and intuitive characterization of utility for the private set union problem. It is also quite surprising that the authors are able to show tightness of their algorithm, especially given that it is significantly simpler than the algorithmic framework in Gopi et al., which involves more complex histogram updating strategies.

In conclusion, I believe this paper would be a good fit for NeurIPS.

---

> ### Author Rebuttal · Authors · 2025-07-30
>
> Thank you for taking the time to review our work. We address your specific points below.
>
> **> For the union set case where k=1, you refer to the sampling-based algorithm from the Cohen et al. paper. I believe there is another algorithm based on thresholding the histogram... I believe the thresholding approach achieves asymptotically similar error bounds... perhaps using the thresholding algorithms as the benchmark would be better?**
>
> We agree that the thresholding approach is very intuitive, and that it achieves asymptotically similar error bounds. But since the strictly optimal (not just asymptotic) reporting probabilities are known in the k=1 case, and since we can actually achieve it under some assumptions (see Theorem 1.3) then we believe that working with the optimal reporting probabilities is a stronger baseline.

---

> ### Comment · Reviewer_wkMV · 2025-08-01
>
> Thanks for your response, will keep score.

---

### Official Review · Reviewer_Y51E · 2025-07-02

**Clarity:** 3
**Significance:** 2
**Originality:** 3
**Rating:** 5
**Confidence:** 2

**Summary:**

This paper investigates the private set union problem when each user may contribute multiple items. The authors introduce a new performance metric called the “utility ratio,” which normalizes expected output size by a dataset-specific upper bound and measures worst-case performance across all datasets. They prove that instance-optimal algorithms exist for the single-contribution case but fail when users can submit multiple items, characterizing how performance degrades as the maximum contribution count grows. For settings with a prior histogram prediction, they design a mechanism that achieves the optimal utility ratio when the prediction is exact and degrades gracefully with prediction error. Both theoretical analyses and empirical evaluations across various settings demonstrate that their algorithms achieve optimal or near-optimal utility ratios.

**Questions:**

For high-k scenarios where per-user contribution counts are large, are there feasible algorithmic improvements or relaxation schemes to mitigate the performance degradation?

What are the computational and communication costs of the mechanisms, and how do they compare empirically to existing methods?

How sensitive is the proposed robust mechanism to different error distributions in the prior histogram prediction?

How does the utility ratio perform across different privacy-utility trade-off points as privacy parameters ($\epsilon$, $\delta$) vary?

**Ethical Concerns:**

["NO or VERY MINOR ethics concerns only"]

**Final Justification:**

The rebuttal has well addressed all of my initial concerns. I think this paper is technically solid, and other reviewers also recognize it as well-structured with significant theoretical contributions to differential privacy. Therefore, I recommend acceptance.

**Limitations:**

Yes

**Quality:**

3

**Strengths And Weaknesses:**

Strengths:

The paper makes significant theoretical contributions, proving tight lower and upper bounds on the utility ratio and constructing matching algorithms. It also extends to scenarios with prior histogram information, offering robust mechanisms with provable guarantees. The work deepens our understanding of differentially private set union with multiple contributions, revealing fundamental limitations and guiding algorithm design. The introduction of the utility ratio provides a unified framework for comparing mechanisms across datasets.

Weaknesses:

While the authors characterize how the worst-case utility ratio deteriorates as per-user contribution bounds grow, they do not propose or evaluate any algorithmic techniques to mitigate this degradation. In extreme multi-item settings, the utility may drop sharply without clear avenues for improvement.

There is little discussion of the runtime complexity or messaging costs of the proposed algorithms. In privacy-constrained settings, these overheads can dominate, especially when each user contributes multiple items, but the paper omits any comparative performance profiling against existing baselines.

The robust mechanism fundamentally assumes access to a reasonably accurate prior histogram of the data distribution. However, the paper does not comprehensively explore how sensitive the overall utility is to different types of prediction errors or misspecified priors.

The work fixes privacy parameters ($\epsilon$, $\delta$) and does not systematically study the privacy–utility frontier. As a result, readers lack guidance on how the utility ratio behaves under tighter privacy budgets or in regimes where sensitivity amplification techniques might apply.

---

> ### Author Rebuttal · Authors · 2025-07-30
>
> Thank you for taking the time to review our work. We address your specific points below.
>
> **> While the authors characterize how the worst-case utility ratio deteriorates as per-user contribution bounds grow, they do not propose or evaluate any algorithmic techniques to mitigate this degradation.**
>
> Our results show that in the worst case, every mechanism is subject to low utility ratios for some privacy regimes and datasets, so mitigations do not exist. However, one interesting direction that we partially address is to explore non-worst-case conditions under which mechanisms can achieve high utility ratios. Our mechanism that makes use of predictions (Section 4) is one example: in cases where a reasonably accurate prediction of the histogram is available, it is possible to do much better than our impossibility results.
>
> **> There is little discussion of the runtime complexity or messaging costs of the proposed algorithms. In privacy-constrained settings, these overheads can dominate, especially when each user contributes multiple items, but the paper omits any comparative performance profiling against existing baselines**
>
> Let $N = \sum_i |B_i|$ denote the sum of bag sizes in an input dataset. The mechanisms discussed in Lemma 3.1, Lemma 3.4, and Theorem 4.1 all run in linear time in $N$ because they essentially build a histogram of the items that appear in the dataset and then do linear time operations on the histogram. The mechanism from Lemma 3.3 is also linear in $N$, since whenever it does not output an empty set, it just outputs the items of a randomly chosen user. Finally, the bi-criteria mechanism from Theorem 3.2 can also be implemented to run in linear time in $N$ with a bit of bookkeeping. Note that the linear dependence on N is unavoidable for any algorithm that goes over every data point. We will clarify this.
>
> **> The robust mechanism fundamentally assumes access to a reasonably accurate prior histogram of the data distribution. However, the paper does not comprehensively explore how sensitive the overall utility is to different types of prediction errors or misspecified priors**
>
> Theorem 4.1 shows that the expected output size of the mechanism that uses predictions degrades with the $\ell_\infty$ distance between the true and predicted dataset histograms. If the distance is $d$, then the mechanism performs as well as the Pi bound on a dataset where every item count is decreased by $d$. Studying robustness with respect to other discrepancy measures is an interesting future direction to explore. We will clarify this in the paper.
>
> **> The work fixes privacy parameters (epsilon, delta) and does not systematically study the privacy-utility frontier. As a result, readers lack guidance on how the utility ratio behaves under tighter privacy budgets or in regimes where sensitivity amplification techniques might apply**
>
> We did not emphasize the privacy-utility frontier, but our results do characterize some interesting regions of the privacy parameter space where utility ratios are bounded in different ways. For example, Lemmas 3.3 and 3.4 show that for fixed $\delta$, utility ratios of 1 are achievable in the limit as $\epsilon$ tends to zero or infinity. On the other hand, for a range of intermediate $\epsilon$ values, our results establish that utility ratios cannot be better than roughly $1/k$. Understanding implications for amplification, composition, etc. would be a nice direction for future research.

---

> > ### Comment · Reviewer_Y51E · 2025-08-01
> >
> > I appreciate the authors’ efforts in the rebuttal and will maintain my positive rating.

---

### Official Review · Reviewer_wdtm · 2025-07-03

**Clarity:** 4
**Significance:** 3
**Originality:** 4
**Rating:** 5
**Confidence:** 3

**Summary:**

This paper studies the problem of differentially private set union: given a collection of sets $ D = (B\_i)\_{i=1}^n $, each contributed by an individual, output a set $ Y $ approximating $ \\bigcup\_{i=1}^n B\_I $ in an $ ( \\varepsilon, \\delta ) $-differentially-private way.
$ Y $ must be a subset of $ \\bigcup\_{i=1}^n B\_I $, and the goal is to make it as large as possible subject to that constraint.

A key parameter is $ k $, an upper bound on the cardinality of each $ B\_i $.

The paper presents new mechanisms for this problem, and also proves impossibility results.

Previous work showed there is a function $ \\Pi(D,\\varepsilon,\\delta) $ which upper bounds the performance of any $ (\\varepsilon,\\delta) $-DP mechanism on a particular dataset $D$, and that when $k=1$ there is a mechanism that achieves this for all datasets $D$.

This paper shows that in contrast, when $k>1$, no mechanism can achieve the $ \\Pi(D,\\varepsilon,\\delta) $ bound for all datasets, even in the "easy" case when $ \\Pi(D,\\varepsilon,\\delta) $ is on the order of the size of the answer.

On the other hand, the paper describes new mechanisms for the $k>1$ regime.
One of them achieves a provable performance guarantee for all datasets (which necessarily is worse than $ \\Pi(D,\\varepsilon,\\delta) $).
The other mechanism is parameterized by an estimate $D$ of the true output, and achieves the limit $ \\Pi(D,\\varepsilon,\\delta) $ if the estimate is exactly correct, with performance decreasing as the quality of the estimate decreases (while always satisfying the privacy requirement, even if the estimate is completely wrong).
It is worth noting that the trivial strategy of always outputting $D$ is not allowed, since the mechanism's output must always be a subset of the true answer.

**Questions:**

- I'm not sure Definition 1.1 has the right citation. I have not checked carefully, but my understanding is Dwork et al's "Calibrating noise to sensitivity in ..." only introduced $\\epsilon$-DP. I have seen the $(\\epsilon,\\delta\)$ version in Definition 1.1 attributed to Dwork, Kenthapadi, McSherry, Mironov, Naor's "Our Data, Ourselves: Privacy via Distributed Noise Generation" in Eurocrypt 2006. Probably that one should be cited. (I skimmed that paper, it looks like they call it "$\\delta$-approximate $\\varepsilon$-indistinguishability", and they seem to be presenting it as an original definition.)

- The definition of $\\Pi(D,\\varepsilon,\\delta)$ on lines 56–60 doesn't say what happens when $\\varepsilon=0$. At least, the constants $c\_\\ell,c\_h$ have $1/\\varepsilon$ in them. The $\\varepsilon=0$ case comes up on line 87, and in the statements of Theorems 2.1 and 2.2, and on line 267.

- Line 83 describes "the bounds rougly mirroring the behavior in Figure 2", but for large $\\varepsilon$, Figure 2 seems to contradict the bound in question, since the utility ratio starts to increase again and eventually reaches 1 as $ \\varepsilon $ increases. I guess it is not really a contradiction because Theorem 1.1's minimum value for $\\delta$ depends on $\\varepsilon$, and so maybe it is not satisfied on the right side of Figure 2. Still, it makes that text on line 83 look untrue. (Or did I misunderstand something here?)

- The second bound in Theorem 1.1 (lines 80-81) has an unspecified dependence on $n$. This is indicated by the subscript $n$ in the $\\tilde{O}\_n$ notation, but it may be unusual or unintuitive enough that it's worth commenting on in the prose: how exactly should one think of this bound? What does it mean? At least, I found it a bit tricky to wrap my head around, since usually $n$ is the increasing variable in asymptotic bounds.

- Line 236: Not strictly necessary, but it would be nice to hear more about what $B\_i$ is. Is there a precise mathematical definition of $B\_i$ in terms of the query $q\_i$ and dataset $D$? Or maybe an example would be appropriate, just to give some intuition?

- Small things:

	- Line 38: "Two datasets $D$ and $D'$ as neighboring..." seems to be be missing something, e.g. "We consider two datasets $D$ and $D'$ as neighbouring...".

	- Line 135: "after removing user $i$": unless I missed it, $i$ isn't defined here. I guess you mean user 1?

**Ethical Concerns:**

["NO or VERY MINOR ethics concerns only"]

**Final Justification:**

I only had relatively minor questions and the authors have addressed them. I'm maintaining my original positive rating.

**Limitations:**

yes

**Quality:**

4

**Strengths And Weaknesses:**

# Strengh: Clear writing

The paper was a pleasure to read. The background and the new results were easy to understand.

# Strength: New mechanisms for private set union

The paper describes two new mechanisms for private set union with $k>1$: a general-purpose one (Theorem 1.2) and one that takes advantage of a prediction of what the set union is likely to include (Theorem 1.3) to get a surprising accuracy guarantee.

# Strength: Impossibility results and mathematical analyses

The paper includes impossibility results that their new mechanisms can be compared to, as well as rigorous mathematical analyses of the new mechanisms.
(The reduction from matrix marginals in the proof of Theorem 2.3 was nice and pleasingly simple.)

Also, there as an interesting use of linear programming to show the best utility ratios acheivable for certain ranges of parameters. (Lines 73-76 and Figure 2.)

# Minor weakness: no empirical evaluation

The paper stands on its own as it is, but it would have been nice to see the method tried out, even on synthetic data.
It's always nice to see an example when understanding the performance of something.

---

> ### Author Rebuttal · Authors · 2025-07-30
>
> Thank you for taking the time to review our work. We address your specific points below.
>
> **> I'm not sure Definition 1.1 has the right citation**
>
> Agreed. This was indeed a mistake and we will update the citation.
>
> **> The definition of $\Pi(D, \epsilon, \delta)$ on lines 56-60 doesn't say what happens when epsilon=0**
>
> In the case when $\epsilon = 0$, we have $\pi(c, \epsilon, \delta) = \min(c \delta, 1)$. We will clarify this in the paper.
>
> **> Line 83 describes "the bounds roughly mirroring the behavior in Figure 2", but for large k, Figure 2 seems to contradict the bound in question, since the utility ratio starts to increase again and eventually reaches 1 as k increases. I guess it is not really a contradiction because Theorem 1.1's minimum value for k depends on epsilon, and so maybe it is not satisfied on the right side of Figure 2. Still, it makes that text on line 83 look untrue. (Or did I misunderstand something here?)**
>
> Your understanding is correct, our impossibility results establish combinations of the privacy parameters $\epsilon$ and $\delta$ under which high utility ratios are impossible. But, for fixed $\delta$, taking $\epsilon$ to zero or infinity allows for a utility ratio of 1 (see Lemmas 3.3 and 3.4), which matches the behavior in the plots.  We intended for the text on line 83 to point out that the curves for each value of k have minimum values near 1/k, the worst-case ratio from the first part of Theorem 1.1. We agree that it is currently confusing and we will revise the text.
>
> **> The second bound in Theorem 1.1 (lines 80-81) has an unspecified dependence on n. This is indicated by the subscript n in the $\tilde O_n$ notation, but it may be unusual or unintuitive enough that it's worth commenting on in the prose: how exactly should one think of this bound? What does it mean? At least, I found it a bit tricky to wrap my head around, since usually n is the increasing variable in asymptotic bounds**
>
> The main focus of our paper is to study how the user contribution limit k impacts the utility ratio, so we emphasize the dependence on k in Theorem 1.1. The full bound is presented in Theorem 2.3, which includes the dependence on n. The bound holds when k is large compared to n and hence we include n as a subscript. We will clarify the notation used in Theorem 1.1 in the revision.
>
> **> Line 236: Not strictly necessary, but it would be nice to hear more about what $B_i$ is. Is there a precise mathematical definition of $B_i$ in terms of the query $q_i$ and dataset $D$? Or maybe an example would be appropriate, just to give some intuition?**
>
> The current text on line 236 uses clashing notation with the rest of the paper which we will address. In the work of Kaplan et al., the query $q_i$ counts the number of users u that match a predicate q and the set of users to remove (denoted by $B_i$ in the current version) contains all users u for which q(u) = 1.
>
> **> Line 38: "Two datasets D and D' as neighboring..." seems to be be missing something, e.g. "We consider two datasets D and D' as neighbouring..."**
>
> Right. We will fix this typo.
>
> **> Line 135: "after removing user i": unless I missed it, i isn't defined here. I guess you mean user 1?**
>
> Right. We will fix this typo.

---

> > ### Comment · Reviewer_wdtm · 2025-08-06
> >
> > Thank you, your responses have addressed my questions. I am maintaining my positive rating.

---

### Decision · Program_Chairs · 2025-09-17

**Decision:**

Accept (spotlight)

**Comment:**

This paper considers the problem of computing a differentially private set union when each user holds at most k items. Past work provided optimal mechanisms for k=1. This work provides theoretical results for k > 1. No mechanism is optimal or can achieve high utility for every dataset. The authors therefore consider the utility ratio, a dataset-normalized utility measure, and prove impossibility results and construct algorithms with strong theoretical guarantees for utility ratio.

Reviewers were unanimously in favor of this paper. They found the problem and the results to be important, with the first substantive results for the k > 1 case and strong impossibility and possibility results, clear writing, and insightful mathematical analysis and proofs. Few or no substantive weaknesses were raised. One reviewer suggested to avoid or qualify the name “private set union” since this has a different meaning in cryptography, which is a useful suggestion. Congratulations on a well received paper.